# ZETA: Leveraging $Z$-order Curves for Efficient Top-$k$ Attention

**Qiuhao Zeng**♠  **Jerry Huang**♡◇  **Peng Lu**♡  **Gezheng Xu**♠
**Boxing Chen**♣  **Charles Ling**♠★  **Boyu Wang**♠★∗
♠University of Western Ontario  ♡Université de Montréal  ◇Mila
♣Noah's Ark Lab  ★Vector Institute

## ABSTRACT

Over recent years, the Transformer has become a fundamental building block for sequence modeling architectures. Yet at its core is the use of self-attention, whose memory and computational cost grow quadratically with the sequence length $N$, rendering it prohibitively expensive for long sequences. A promising approach is top-$k$ attention, which selects only the $k$ most relevant tokens and achieves performance comparable to vanilla self-attention while significantly reducing space and computational demands. However, causal masks require the current query token to only attend to past tokens, preventing existing top-$k$ attention method from efficiently searching for the most relevant tokens in parallel, thereby limiting training efficiency. In this work, we propose ZETA, leveraging **Z**-Order Curves for **E**fficient **T**op-$k$ **A**ttention, to enable parallel querying of past tokens for entire sequences. We first theoretically show that the choice of key and query dimensions involves a trade-off between the curse of dimensionality and the preservation of relative distances after projection. In light of this insight, we propose reducing the dimensionality of keys and queries in contrast to values and further leverage $Z$-order curves to map low-dimensional keys and queries into *one*-dimensional space, which permits parallel sorting, thereby largely improving the efficiency for top-$k$ token selection. Experimental results demonstrate that ZETA matches the performance of standard attention on the synthetic MULTI-QUERY ASSOCIATIVE RECALL task and outperforms attention and its variants on LONG RANGE ARENA and WIKITEXT-103 language modeling.

## 1 INTRODUCTION

Transformers (Vaswani et al., 2017) have become indispensable for sequence modeling across various domains (OpenAI et al., 2024; Zeng et al., 2024a;b; 2023; Fang et al., 2022; 2025), including natural language processing (NLP) (Devlin et al., 2019; Brown et al., 2020; OpenAI et al., 2024; Jiang et al., 2024), computer vision (Dosovitskiy et al., 2021; Ramesh et al., 2021; Brooks et al., 2024), etc. The foundation of Transformer models is the self-attention mechanism. This mechanism (Bahdanau et al., 2015), inspired by recurrent neural networks (RNNs) and their ability to construct representations from all elements in a sequence, has revolutionized numerous fields, enabling breakthroughs in tasks such as language modeling (Radford et al., 2019), machine translation (Ott et al., 2018), text generation (Brown et al., 2020), image classification (Touvron et al., 2021) and video generation (Brooks et al., 2024). However, self-attention has a quadratic complexity in both memory and computation as the sequence length $N$ increases, which presents a significant challenge when scaling to long sequences (Child et al., 2019; Beltagy et al., 2020). This makes the direct application of self-attention in large-scale problems computationally prohibitive for many real-world applications, particularly when long sequences are involved (Tay et al., 2021).

Recent advances have explored strategies to mitigate the inefficiencies of vanilla self-attention. One such approach is top-$k$ attention, which focuses computation on a subset of the most relevant tokens, significantly reducing memory and computation costs while maintaining competitive performance (Kitaev et al., 2020; Gupta et al., 2021; Bertsch et al., 2023; Mao et al., 2024). However,

---

∗Corresponding author: Boyu Wang.

existing top-$k$ attention methods (Kitaev et al., 2020; Zhuoran et al., 2021) typically apply causal masking after selecting the top-$k$ tokens, causing earlier tokens ($i \ll N$) to often attend to nothing as their top-$k$ relevant tokens may include future tokens that are masked out by the causal mask. Alternatively, Mao et al. (2024) process input tokens by token to exclude masked-out keys from the $k$-nearest neighbors search, preventing future tokens from influencing currently generated ones. Consequently, with causal masks, current top-$k$ attention approaches fail to fully leverage the parallel computation abilities of modern accelerators, limiting their efficiency in long-sequence modeling.

To overcome the limitations of existing top-$k$ attention methods, we introduce ZETA, a novel model designed to search for the top-$k$ tokens within chunked one-dimensional (one-dimensional) sorted key sequences projected via $Z$-order curves. Specifically, our approach strikes a balance between mitigating the "curse of dimensionality" and preserving the relative distance of token represntations after projection by carefully selecting a lower dimensionality for keys and queries in contrast to values. This reduction allows the query and key to map to a one-dimensional space using $Z$-order curves, preserving proximity. As a result, ZETA efficiently performs top-$k$ token selection in parallel within this one-dimensional space with accelerators. Additionally, since the aforementioned top-$k$ search is based on the Euclidean metric for low dimensional data, directly applying the traditional dot-product based softmax function is not appropriate. To address this, we propose an Adaptive Cauchy-Softmax mechanism that replaces the exponential function in the attention operation with a trainable Cauchy kernel (Billingsley, 1986). This enables dynamic adjustment of receptive fields across layers, providing greater flexibility in capturing both short and long-range dependencies.

Extensive empirical evaluations show that ZETA matches the performance of standard self-attention on the Associative Recall task and consistently outperforms existing attention variants on the Long-Range Arena (LRA) and WikiText-103 datasets. We summarize our key contributions as follows:

- **Efficient Parallel Top-$k$ Attention**: We introduce ZETA, a novel model that enables top-$k$ attention to operate in parallel across entire sequences, significantly improving training and inference efficiency with a time complexity of $\mathcal{O}(N \log N)$.

- **Dimensionality Selection for Key and Query Pairs**: We theoretically show that the dimensionality of keys and queries decides the trade-off between the curse of dimensionality and the preservation of relative distances for keys and queries.

- **$Z$-order Curve Integration**: By leveraging $Z$-order curves, we enable efficient top-$k$ token selection in one-dimensional space, allowing the use of parallel sorting algorithms on GPUs for faster attention computation.

- **Adaptive Cauchy-Softmax Mechanism**: We introduce Adaptive Cauchy-Softmax, a Softmax variant with trainable parameters based on the Cauchy kernel, dynamically adjusting receptive fields to enhance attention's flexibility.

## 2 RELATED WORKS

**Efficient Transformer** The Transformer architecture (Vaswani et al., 2017) is foundational for sequence modeling, but its quadratic complexity limits efficiency with long sequences. Various efficient variants (Tay et al., 2022; 2020; Chen et al., 2021; Qin et al., 2022b; Zhang et al., 2024) have been proposed as alternatives, mainly categorized into sparse, low-rank, and linear transformers. Sparse transformers, such as BigBird (Zaheer et al., 2020) and Longformer (Beltagy et al., 2020), restrict attention to local windows or global tokens to achieve linear complexity. SparseAxial (Ho et al., 2020) further enhances this by combining sparse attention with axial mechanisms for high-dimensional inputs. Reformer (Kitaev et al., 2020) locality-sensitive hashing to handle variable-length sequences efficiently. Low-rank transformers like Linformer (Wang et al., 2020) reduce the attention matrix to a lower-dimensional space, reducing memory and computation costs. Linear transformers such as Performer (Choromanski et al., 2021) use kernel-based approximations for linear-time complexity, while Nyströmformer (Xiong et al., 2021) leverages Nyström decomposition for near-linear performance.

**Top-$k$ Attention** (Gupta et al., 2021) falls under the category of sparse attention, reducing attention complexity by selecting only the top-$k$ most relevant tokens at each layer, thereby focusing computational resources on the most critical interactions. Unlimiformer (Bertsch et al., 2023) enables transformers to handle arbitrarily long sequences by chunking inputs and using a retrieval

mechanism to attend to relevant past contexts. Similarly, IceFormer (Mao et al., 2024) improves transformer efficiency by integrating a $k$-nearest-neighbor (KNN) search mechanism that focuses on the KNN results as the most relevant tokens during inference, bypassing the need to compute the full attention matrix. However, with causal masks, these approaches can not compute the outputs of a long sequences in parallel, making them less efficient for training models from scratch by not fully exploiting the parallel computing power of GPUs. In contrast, ZETA performs KNN-based searches for relevant tokens in parallel across the entire sequence on GPUs using chunking techniques, enabling efficient training and inference with a time and space complexity of $\mathcal{O}(N \log N)$.

## 3 METHODOLOGY

### 3.1 PRELIMINARIES

**Attention** has proven to be a fundamental building block in modern deep learning, particularly in natural language processing and sequence modeling tasks. It allows a model to focus on specific parts of the input sequence, thereby capturing dependencies within the sequence more effectively. In the standard formulation of attention (Vaswani et al., 2017), given a set of queries $\boldsymbol{Q} \in \mathbb{R}^{N \times d_Q}$, keys $\boldsymbol{K} \in \mathbb{R}^{N \times d_K}$, and values $\boldsymbol{V} \in \mathbb{R}^{N \times d_V}$, the attention scores are computed as

$$\text{Attention}(\boldsymbol{Q}, \boldsymbol{K}, \boldsymbol{V}) = \text{softmax}\left(\boldsymbol{Q}\boldsymbol{K}^T / \sqrt{d_K}\right)\boldsymbol{V}, \tag{1}$$

where $d_K$, $d_Q$ and $d_V$ is the dimension of the keys, queries and values, respectively. It is common to use the same value for all three.

**Top-$k$ Attention** approaches aim to enhance the efficiency and flexibility of traditional self-attention mechanisms by focusing attention to only the most relevant tokens in a sequence. Both methods reduce the computational overhead associated with self-attention, especially for long sequences, by selectively attending to the most important tokens rather than computing attention scores across the entire sequence. This process lowers the computational complexity from $\mathcal{O}(N^2)$ in traditional self-attention to $\mathcal{O}(N \cdot k)$, where $N$ is the sequence length and $k$ is the number of selected tokens. The top-$k$ highest scores are selected for each query:

$$I_q = \left\{i \mid \boldsymbol{q}\boldsymbol{K}_i^T / \sqrt{d_K} \geq \tau_k\right\} \tag{2}$$

where $I_q$ represents the set of indices corresponding to the top-$k$ highest attention scores for the query $q$, $\tau_k$ is the threshold defined as the $k$-th highest attention score. The attention is then restricted to the tokens in this subset, reducing the number of operations required for long sequences. Specifically, the attention score for the top-$k$ tokens is recalculated using the self-attention mechanism but limited to the selected indices:

$$\text{Attention}_{\text{top-}k}(\boldsymbol{q}, \boldsymbol{K}, \boldsymbol{V}) = \sum_{i \in I_q} \text{softmax}\left(\boldsymbol{q}\boldsymbol{K}_i^T / \sqrt{d_K}\right)\boldsymbol{V}_i \tag{3}$$

where $\boldsymbol{q}$ denotes the query vector, $\boldsymbol{K}_i$ and $\boldsymbol{V}_i$ denote the key and value vectors corresponding to the top-$k$ relevant tokens, respectively.

$Z$**-order Curves** (Dugundji, 1966), also known as Morton codes, provide a way to map multi-dimensional data into a one-dimensional space while preserving locality, whereas other dimensionality reduction methods (Abdi & Williams, 2010; McInnes et al., 2018) are not designed for mapping data to a 1D space. This approach is valuable in tasks that require efficient spatial indexing or key-query matching, such as attention. By maintaining the relative proximity of data points after projection, $Z$-order curves ensure that points that are close together in the original multi-dimensional space remain close in the projected one-dimensional space.

The $Z$-order curve interleaves the binary representations of each coordinate in a multi-dimensional point. For a point in the $d$-dimensional space with coordinates $\boldsymbol{x} = (x_1, x_2, \ldots, x_d)$, where each $\boldsymbol{x}_i$ is a binary number, the $Z$-order curve computes a scalar value $Z$ by interleaving the bits of each coordinate. Given the binary representation of $x_i$ as $b_{i1}b_{i2}\ldots b_{in}$, the $Z$-order curve is expressed as:

$$Z = b_{11}b_{21}\ldots b_{d1}b_{12}b_{22}\ldots b_{d2}\ldots b_{1n}b_{2n}\ldots b_{dn} \tag{4}$$

where $n$ refers to the number of bits used to represent each coordinate $x_i$ in its binary form. Through this interleaving of bits, the $Z$-order curve creates a scalar value that allows efficient sorting or indexing of points while approximately maintaining their original spatial relationships.

The primary advantage of $Z$-order curves is their ability to preserve locality. In other words, nearby points in the original multi-dimensional space have similar $Z$-values in the projected one-dimensional space. This property enables efficient search and selection processes in attention mechanisms or spatial indexing, where key-query pairs can be processed more efficiently in the one-dimensional space without significantly losing the locality information from the higher-dimensional space. $Z$-order curves are designed to preserve locality, and hence not suitable for dot-product similarity measures, which does not reflect locality.

## 3.2 Searching for the Top-$k$ Attended Tokens in One-Dimensional Space

Since we project key and query vectors into a one-dimensional space using $Z$-order curves, using a large $d_K$ can still distort locality (as shown in Figure 3) and compromise the preservation of relative distances. Thus we ask whether $d_K$ can be reduced in a way such that even after mapping to one dimension, relative distances between tokens are maintained. Importantly, the key and query dimensions $d_K$ and $d_Q$ do not have to match the dimension of the values $d_V$. This is because $d_V$ should remain large to capture more semantic information, as seen with Gaussian distributions, where higher dimensionality increases the measure of information entropy (Cover & Thomas, 2006). Hence, as long as the relative distances between queries and keys are preserved, $d_K$ and $d_Q$ can be reduced. The following theoretical analysis provides insights into the selection of $d_K$.

### 3.2.1 Theoretical Analysis on $d_K$

We first introduce the Johnson–Lindenstrauss Lemma (Johnson et al., 1986), which states that data in high-dimensional space can be projected into a much lower-dimensional subspace using random projections while approximately preserving the pairwise distances between the points. Since random projections can preserve locality, this provides justification for setting a smaller $d_K$ with trainable projection functions for keys and queries, which could also preserve locality.

**Lemma 3.1.** *(Johnson–Lindenstrauss Lemma) For any $0 < \epsilon < 1$ and any integer $m$, let $d$ be a positive integer such that $d = \Omega(\frac{\ln m}{\epsilon^2})$. Then for any set $x$ of $m$ points in $\mathbb{R}^D$, there exists a map $f : \mathbb{R}^D \to \mathbb{R}^d$ such that for all $x_i, x_j \in \mathcal{X}$,*

$$(1 - \epsilon)\|x_i - x_j\|^2 \leq \|f(x_i) - f(x_j)\|^2 \leq (1 + \epsilon)\|x_i - x_j\|^2 \tag{5}$$

The following assumption then provides a mathematical depiction that attention weights are constrained within an $m$-dimensional simplex, and the learnable similarity function $\Gamma$ outputs the attention scores, ensuring the most relevant tokens are emphasized during the information aggregation process. This reflects the primary goal of attention: to aggregate critical information for more accurate predictions.

**Assumption 3.2.** Let $\alpha \in \Delta_{m-1}$ be an element of the $m$-dimensional simplex, defined as $\Delta_{m-1} \triangleq \{\alpha \in \mathbb{R}^m \mid \alpha_i \geq 0, \sum_{i=1}^{m} \alpha_i = 1\}$. Assume that $h_{\text{attn}}$ equipped with $\alpha$ can achieve an optimal learnable similarity critic function $\Gamma$, where the attention scores are given by $\alpha = \text{softmax}(\Gamma(f_k(x_i), f_q(x)))$, such that $\Gamma$ is trained to be optimal to have the minimal expected risk: $\min_\alpha \|h_{\text{attn}}(x, S_x; \alpha) - y\|$, where $h_{\text{attn}}$ denotes the attention-based hypothesis, $x$ is the input, and $S_x$ is the context.

Theorem 3.3 highlights the importance of choosing $d_K$ carefully, as it controls for a trade-off between locality preservation and the curse of dimensionality. Larger $d_K$ allow for more detailed feature capture at the cost of the high-dimensional curse, leading to increased complexity. On the other hand, a smaller $d_K$ loses locality between tokens, which is crucial for efficient query. The bounds provided give valuable insights into the underlying mechanisms of attention and can guide future designs of more efficient attention models. For simplicity, we assume WLOG that keys and queries share the same projection functions, as Kitaev et al. (2020).

**Theorem 3.3.** *Let $\mathcal{X} \in \mathbb{R}^d$, $\mathcal{Y} \in \mathbb{R}^D$, and $\mathcal{D}$ be a distribution over $\mathcal{X} \times \mathcal{Y}$ for which the conditional probability function, $h : \mathbb{R}^d \to \mathbb{R}^D$, is a $l$-Lipschitz function. Let $h$ denote a hypothesis, and $h_{\text{attn}}$*

*denote the one-layer attention model to aggregate the predictions of a sample set $S \sim \mathcal{D}^m$ to predict for another i.i.d sample $x$. Specifically, here we assume the same linear map for a key mapping $f_k$ and a query mapping $f_q$ as $f : \mathbb{R}^d \to [-B, B]^{d_K}$ where $B$ is the bound of projection of $x$ by $f$, and we assume the value mapping $f_v$ be the Bayes optimal rule $h^* = \mathbb{E}[Y|X = x]$, which is the hypothesis that minimizes $L_{\mathcal{D}}(h)$ over all functions. Then, the expected risk of the regression tasks $L_{\mathcal{D}}(h) = \mathbb{E}_{(x,y)\sim\mathcal{D}}\|h(x) - y\|$ with $h$ as $h_{\mathrm{attn}}$ can be upper bounded by*

$$\underset{S \sim \mathcal{D}^m}{\mathbb{E}} \left[ L_{\mathcal{D}}(h_{\mathrm{attn}}) \right] \leq L_{\mathcal{D}}(h^*) + \frac{4lc\sqrt{d_K}Bm^{-1/(d_K+1)}}{\sqrt{1 - \sqrt{\frac{C\ln m}{d_K}}}}$$

This indicates that $d_K$ should be carefully chosen rather than simply being set as equal to $d_V$ (as is common practice). Empirically, we show that the dimension of query and key can decreased without degrading performance, more specifically in Figure 2b in Section 4.2 which we later discuss.

### 3.2.2    TOP-$k$ SEARCH IN ONE DIMENSION

As the sequence length becomes extremely large, iterating through the entire context history to search for the top $k$ tokens becomes infeasible. Ideally, searching should be efficient and aim to achieve the optimal time complexity of $\mathcal{O}(N \log N)$, similar to general sorting problems with arbitrary inputs. To achieve this, we map both keys and queries into a one-dimensional space using the $Z$-order curve and sort them with a sorting operation that can be executed in parallel on an accelerator, e.g. the `torch.sort` operator in PyTorch (Paszke et al., 2019). The insertion position of a query in the key sequence can then be found using a binary search (e.g. with `torch.searchsorted`), allowing us to retrieve the top-$k$ attended tokens using a window centered on the insertion position.

Specifically, the key and query dimensions are set to be significantly smaller than that of the values, i.e. $d_K = d_Q \ll d_V$. While values need a high dimensionality to carry rich semantic information, keys and queries primarily serve to preserve relative distances, which can be achieved with much lower dimensionality as argued by Lemma 3.1. To facilitate the fast retrieval of queries from keys, we leverage sorting, which can be efficiently parallelized, after mapping queries and keys into one-dimensional space via the $Z$-order curve. We define $\boldsymbol{Q}, \boldsymbol{K} \in \mathbb{R}^{B \times N \times d_K}$, where $B$ is the batch size and $N$ is the sequence length. The $Z$-order transformation is applied as follows:

$$\boldsymbol{Q}_z = Z\text{-order}(\boldsymbol{Q}), \quad \boldsymbol{K}_z = Z\text{-order}(\boldsymbol{K})$$

where $\boldsymbol{Q}_z$ and $\boldsymbol{K}_z$ are the one-dimensional representations of $\boldsymbol{Q}$ and $\boldsymbol{K}$, respectively.

With causal masks, directly collecting the top-$k$ tokens from the entire sequence is not guaranteed to have plenty of tokens to make inference. For instance, for a query at position 32 in a sequence of length 2048, selecting the top 16 tokens from the entire sequence followed by causal masking would leave approximately $\frac{32}{2048} \times 16 = \frac{1}{4}$ tokens to attend to, effectively leaving no tokens available for the query. Consequently, the query at this position would not attend to anything, rendering top-$k$ attention ineffective. To enable parallel $k$-nearest neighbors ($k$NN) searching considering causal masks, we first sort the $Z$-order keys and divide them into chunks. For query $i$ in the $m$-th chunk (where $m = \lfloor i/M \rfloor$ and $M$ is the chunk size), we restrict search to the first $m$ chunks, indexing the original unsorted keys from 0 to $m \times M - 1$ in the sorted list. This ensures that future keys $j \in \{j : j > m \times M\}$ are excluded, in accordance with the causal mask requirements. This process is performed in parallel for every query.

Next, we perform the nearest neighbor search in these one-dimensional $Z$-order spaces. For each query, the insertion position is first found using a binary search and followed by selecting the nearest keys using a window of size $K$ centered around the insertion point to collect top-$k$ tokens as $I_q$, denoting the indices of the $k$-nearest neighbors. This ensures efficient and accurate retrieval while maintaining the constraints imposed by causal masking.

### 3.3    ADAPTIVE CAUCHY-SOFTMAX

Top-$k$ token searching relies on a similarity metric between data points, and we prefer the Euclidean metric for top-$k$ attention with small $d_K$ for two main reasons. First, as illustrated in Figure 1, Euclidean distance is more effective for low-dimensional data in top-$k$

methods: it reliably identifies the correct class in one-dimensional classification tasks, leading to accurate predictions, whereas the dot product can be misleading (samples in the misclassified area will be classified as "+1" using the dot-product metric). Second, $k$-NN search is typically based on the Euclidean metric, while using the dot-product requires normalization that loses token magnitudes.

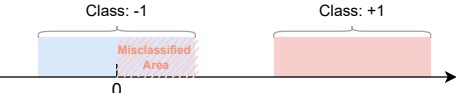

To better align with the Euclidean measure for low-dimensional representations, we propose the Adaptive Cauchy Softmax function, to replace the exponential function in traditional attention mechanisms. The Cauchy kernel, with its heavier tails, ensures

Figure 1: Illustration of attention using Euclidean distance vs. dot product. Euclidean distance correctly classifies points into classes $\pm 1$, while the dot product leads to a misclassified area.

that distant tokens retain influence, overcoming the limitations of the exponential function, which suppresses distant tokens (Shen et al., 2023). This allows the attention mechanism to capture both local and global dependencies, with the shape of the kernel determining how key vectors influence the query. Specifically, the Adaptive Cauchy-Softmax between a query vector $\boldsymbol{q}$ and keys $\boldsymbol{K}$ is computed as:

$$\text{softmax}_c(\boldsymbol{q}, \boldsymbol{K}) = \frac{\frac{\gamma}{\pi} \left[\|\boldsymbol{q} - \boldsymbol{K}_i\|^2 + \gamma^2\right]^{-1}}{\sum_{j \in I_q} \frac{\gamma}{\pi} \left[\|\boldsymbol{q} - \boldsymbol{K}_j\|^2 + \gamma^2\right]^{-1}} = \frac{\left[\|\boldsymbol{q} - \boldsymbol{K}_i\|^2 + \gamma^2\right]^{-1}}{\sum_{j \in I_q} \left[\|\boldsymbol{q} - \boldsymbol{K}_j\|^2 + \gamma^2\right]^{-1}} \quad (6)$$

where $\gamma$ is a trainable parameter that controls the shape of the distribution. By training a task-specific $\gamma$ for each attention layer, the model adjusts receptive fields dynamically. We define $\gamma^2$ as the output of a sigmoid function applied to a trainable parameter, to ensure a range of $[0, 1]$, with smaller values sharpening attention and improving focus on relevant inputs (Qin et al., 2022a; Zhang et al., 2024), while larger values allow for smoother attention. The adaptive Cauchy softmax effectively handles long-range dependencies, preventing entropy collapse (Zhai et al., 2023) or explosion (Zhang et al., 2024) and adaptively balancing attention across the sequence.

### 3.4 SPARSE ATTENTION WITH $Z$-ORDER CURVE FOR EFFICIENT kNN RETRIEVAL

Instead of calculating full attention scores, which is computationally expensive and memory intensive, we compute sparse attention scores by leveraging the $Z$-order curve for efficient nearest neighbor retrieval. ZETA is then computed as below, according to Equation 6:

$$\text{Attention}_{\text{ZETA}}(\boldsymbol{Q}, \boldsymbol{K}, \boldsymbol{V}) = \sum_{i \in I_q} \text{softmax}_c\left(\boldsymbol{Q}, \boldsymbol{K}_i\right) \boldsymbol{V}_i \quad (7)$$

where $\boldsymbol{K}_i$ and $\boldsymbol{V}_i$ are the corresponding keys and values for the indices $i \in I_q$. As a result of the sparsity of top-$k$, most of the tokens will not join the predictions, which stops the gradient back-propagated through the low-probabilities tokens and fails to leverage this current prediction's information. We append the mean vector of the history tokens to the top-$k$ tokens matrix using cumsum function in $I_q$, to keep the attention from assigning zero probability, which can be regarded as smoothing in n-gram language model (Jurafsky & Martin, 2024).

## 4 EXPERIMENTAL RESULTS

We evaluate ZETA's performance on several aspects: ZETA's ability to solve the synthetic MULTI-QUERY ASSOCIATIVE RECALL (MQAR) task (Arora et al., 2024a), long sequence modeling ability on the LONG RANGE ARENA (LRA) benchmark and auto-regressive language modeling on WIKITEXT-103. Then we conduct extensive analysis experiments: an ablation study examining the influence of dimensionality on attention model performance (Section 4.2), ablations on various Euclidean-based Softmax operators (Section 4.3), the empirical results of locality preservation using $Z$-order curves (Section 4.4) and an ablation study over the number of $k$ in ZETA (Section 4.5).

### 4.1 EMPIRICAL VALIDATION

**Associative Recall.** Associative recall tasks (Arora et al., 2024a) have been popular for testing the ability of language models to look up information in their context. Broadly, they involve feeding

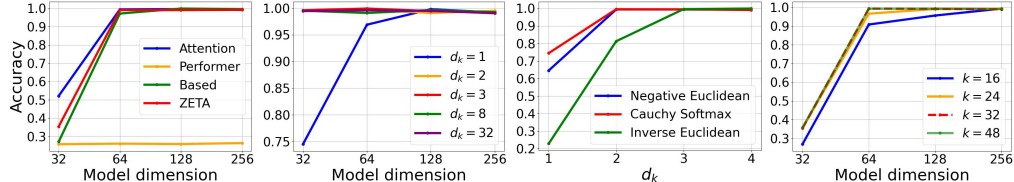

Figure 2: Experiments on Associative Recall: (a) Model Accuracy (b) Performance of Transformer with varying $d_K$ across different model dimensions; even with low $d_K$, the model achieves near-perfect performance (c) Comparison of different Euclidean-based Softmax operators across varying key-query dimensions $d_K$ (d) Ablation on $k$ in ZETA.

auto-regressive model pairs of key-value associations and then prompting the model to produce the correct completion upon being shown a previously seen key. The MULTI-QUERY ASSOCIATIVE RECALL (MQAR) task is a particular formulation of this task that requires the model to memorize multiple associations (Arora et al., 2024a). We evaluate the performance of various models on the Associative Recall task, a classical sequence-to-sequence task that requires the model to recall the first token associated with a target token after processing a long sequence. The task is tests the ability of models to capture long-range dependencies and to maintain information over time.

We compare the performance of four models: a vanilla Transformer, Performer (Choromanski et al., 2021), BASED (Arora et al., 2024b), and ZETA, with different model dimensions (32, 64, 128, and 256). As illustrated in Figure 2a, accuracy increases as the model dimension grows. Attention and Based models show strong performance with higher dimensions, achieving nearly perfect accuracy for dimensions larger than 64. ZETAfollows a similar trend and achieves competitive performance, especially for larger model dimensions, with perfect accuracy at dimension 256. In contrast, the Performer struggles, showing significantly lower accuracy across all dimensions.

**Long Range Arena (LRA).** The LONG RANGE ARENA (LRA) benchmark (Tay et al., 2021) is a comprehensive suite designed to evaluate the performance of models on long sequence tasks. It includes tasks that span across multiple domains, such as natural language processing, image classification, and mathematical reasoning. LRA focuses on sequence classification, challenging models to efficiently process longer input sequences while capturing long-range dependencies, providing an ideal testbed for Transformer models and their efficient variants.

LRA consists of five key tasks: LISTOPS, TEXT, RETRIEVAL, IMAGE, and PATHFINDER. Each task evaluates different aspects of long-range dependency handling such as the ability to handle mathematical reasoning tasks on long sequences of operations (LISTOPS), capture dependencies over long textual inputs (TEXT), retrieve relevant elements from a long sequence (RETRIEVAL) and capture spatial dependencies (IMAGE). PATHFINDER presents a difficult problem where models must distinguish between connected and disconnected paths within maze-like patterns. A modified version of PATHFINDER, called PATHFINDER-X, is also included where the patterns are presented in a larger image (256×256 compared to 32×32) but has yet to be solved by existing attention-based methods.

Table 1: Test perplexity (lower is better) on WIKITEXT-103.

| Model | Params | Test PPL |
|---|---|---|
| Vanilla Transformer | 125M | 26.2 |
| Performer | 125M | 26.8 |
| Reformer | 125M | 25.6 |
| AFT-conv | 125M | 28.2 |
| Linear Transformer | 125M | 30.2 |
| RFA-Gaussian | 125M | 27.5 |
| CosFormer | 125M | **23.1** |
| ZETA | 124M | 26.3 |

We evaluate various Transformer-based models, including several linear and efficient variants, trained from scratch on the LRA sequence classification tasks. For each model, we adopt the same hyperparameter settings provided by the official LRA benchmark (Tay et al., 2021) to ensure a fair comparison. Results are summarized in Table 2, which compares the performance of the models across all five tasks, along with their average accuracy, showing that ZETA significantly outperforms other attention-based models.

Table 2: Results of the Transformer and various variants on LRA. We consistently outperform the next closest competitor (Zhu & Soricut, 2021).

| Model | ListOps | Text | Retrieval | Image | Pathfinder | Average |
|---|---|---|---|---|---|---|
| Transformer | 36.37 | 64.27 | 57.46 | 42.44 | 71.40 | 54.39 |
| Reformer | 37.27 | 56.10 | 53.40 | 38.07 | 68.50 | 50.67 |
| Sparse Trans | 17.07 | 63.58 | 59.59 | 44.24 | 71.71 | 51.24 |
| Sinkhorn Trans | 33.67 | 61.20 | 53.83 | 41.23 | 67.45 | 51.29 |
| Linformer | 35.70 | 53.94 | 52.27 | 38.56 | 76.34 | 51.36 |
| BigBird | 36.05 | 64.02 | 59.29 | 40.83 | 74.87 | 55.01 |
| Linear Trans. | 16.13 | 65.90 | 53.09 | 43.40 | 75.30 | 50.76 |
| Performer | 18.01 | 65.40 | 53.82 | 42.77 | **77.05** | 51.41 |
| Nyströmformer | 41.28 | 58.38 | 65.40 | 37.54 | 71.76 | 54.87 |
| H-Transformer-1D | **49.53** | **78.69** | 63.99 | 46.05 | 68.78 | 61.41 |
| Top-$k$ Attention | 38.12 | 63.72 | 59.14 | $\times$ | $\times$ | 53.66 |
| IceFormer | 41.53 | 60.01 | 66.02 | 40.46 | 74.42 | 56.49 |
| cosFormer | 37.90 | 63.41 | 61.36 | 43.17 | 70.33 | 55.23 |
| Skyformer | 39.25 | 64.70 | 82.06 | 40.77 | 70.73 | 59.50 |
| Hedgehog | 37.15 | 64.60 | **82.24** | 40.15 | 74.16 | 59.66 |
| ZETA | 42.52 | 64.52 | 77.92 | **64.39** | 68.20 | **63.51** |

**Autoregressive Language Modeling.** Furthermore, we evaluate several models on the WIKITEXT-103 (Merity et al., 2017), a widely used benchmark for language modeling containing over 100 million tokens extracted from high-quality Wikipedia articles characterized by a large vocabulary and long-range dependencies. This makes it a challenging benchmark for testing a model's ability to predict the next token in a sequence. We use perplexity (PPL) as the primary evaluation metric, where lower scores indicate a better ability to capture the sequential structure within natural language text. Table 1 shows a vanilla Transformer[1] to achieve a test perplexity of 26.2. However, linear approximation models such as the Linear Transformer (Qin et al., 2022a) struggle to compete, with higher perplexity values of 30.2 on the test set. The table further compares several other models, including efficient attention mechanisms like Performer (Choromanski et al., 2021), Reformer (Kitaev et al., 2020), and CosFormer (Qin et al., 2022b). Notably, CosFormer achieves the lowest perplexity on the test set with a score of 23.1, outperforming all other models. Reformer also shows competitive results, achieving a perplexity of 25.6, improving on the Vanilla Transformer. ZETA achieves a perplexity of 26.3, comparable to the Vanilla Transformer.

The results highlight the trade-offs between using conventional transformers, linear transformers, and models adopting approximate attention mechanisms like ZETA. It reinforces the importance of balancing computational efficiency with model performance, particularly in the context of long-sequence language modeling tasks, especially as the information necessary to solve a task becomes sparsely located within long contexts.

## 4.2 EFFECT OF VARYING $d_K$ ON ASSOCIATIVE RECALL TASK

We next evaluate the effect of different key-query dimensions $d_K$ on the Transformer model's performance for MQAR. The model dimensions are varied between $\{32, 64, 128, 256\}$ while adjusting $d_K$ to values $\{1, 2, 3, 8, 32\}$. As shown in Figure 2b, the performance remains near-perfect even with low $d_K$ values, such as $d_K = 2$. The model achieves close to 100% accuracy across all model dimensions, except for the smallest dimension ($d_K = 1$), where performance slightly drops for lower model dimensions. This demonstrates that the Transformer is capable of handling long-range dependencies in the Associative Recall task, even with relatively low key-query dimensionality.

These results suggest that reducing $d_K$ does not significantly impair the model's ability to recall information in sequence tasks. In fact, maintaining a low $d_K$ can provide computational savings without sacrificing performance, especially when model dimensions are large. This indicates that while random projections—such as those used in the Johnson-Lindenstrauss Lemma—approximately preserve distances, trainable projection networks $f_k$ and $f_q$ can better adapt to task-specific data and more effectively retain locality even with a low $d_K$. For instance, by setting $d_K$ as low as 3, we

---

[1]We use an auto-regressive Transformer based on Biderman et al. (2023) for comparision.

reduce it from the typical head dimension (normally 32, i.e. feature dimension as 512 with 8 heads). We can further mitigate information loss by configuring $f_k$ and $f_q$ as two-layer neural networks rather than single-layer ones.

### 4.3 Performance of Euclidean-Based Softmax Operators

We further evaluate the performance of transformers with various Euclidean-based Softmax operators on MQAR. Specifically, we compare Negative Euclidean, Cauchy Softmax (our proposed method), and Inverse Euclidean operators. The goal of this experiment is to test how these different formulations of Softmax handle varying key-query dimensions $d_K$ in terms of accuracy.

As shown in Figure 2c, the proposed Cauchy Softmax consistently outperforms the other operators across all values of $d_K$. It achieves near-perfect accuracy for $d_K \geq 2$, whereas Negative Euclidean shows a drop in accuracy for lower $d_K$ values. Inverse Euclidean, while performing comparably at higher dimensions, struggles significantly at lower values of $d_K$ (e.g., $d_K = 1$).

These results highlight the advantage of using the Cauchy distribution for a smaller $d_K$, as it allows for better handling of long-range dependencies and achieves more stable performance across various key-query dimensions. The heavier tails of the Cauchy distribution enable distant tokens to retain non-negligible influence, which is crucial for tasks like Associative Recall where long-range token relationships are important.

### 4.4 Locality Preservation after $Z$-order Curve Projection

Next, we evaluate how well $Z$-order curve projections preserve locality across different dimensions and sample sizes. Specifically, we test the locality preservation by measuring the overlap between the top-64 nearest neighbors before and after projection, with sample sizes $N \in \{512, 1024, 2048\}$.

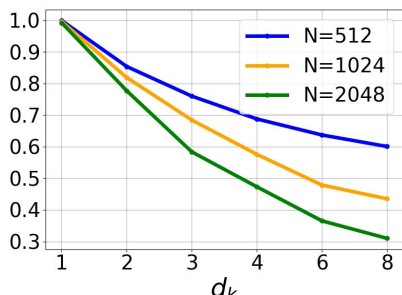

Figure 3 shows the relationship between locality preservation and the dimensionality $d_K$. As $d_K$ increases, the overlap between the top-64 nearest neighbors diminishes for all sample sizes, indicating a decrease in locality preservation. Lower $d_K$ values exhibit a higher level of locality preservation across all sample sizes. However, for larger sample sizes, such as $N = 2048$, the drop in locality preservation is more pronounced as the dimensionality increases. We select $d_K = 3$ for ZETA.

Figure 3: The effect of dimensionality reduction before and after $Z$-order curves projection on locality preservation for different sample sizes.

These results highlight the importance of choosing an appropriate $d_K$ for maintaining locality, especially for larger datasets where higher dimensionality can lead to distortions in spatial proximity after a projection.

### 4.5 The Effects of $k$ in ZETA

As an ablation study, we explore the effect of varying $k$ in the attention mechanism of ZETA. The goal of this experiment is to analyze how different values of $k$ influence the model's performance on the associative recall task across different model dimensions.

Figure 2d shows ZETA to achieve near-perfect accuracy across all model dimensions (32, 64, 128, and 256) for different values of $k$ ranging from 16 to 48. In most of our experiments, we set $k = 32$, as it provides a good balance between performance and computational efficiency. Interestingly, there is little variation in accuracy between different values of $k$, indicating that ZETA is robust to changes in this parameter.

### 4.6 Efficiency Benchmarking

In order to better understand the effectiveness of ZETA, we further conduct an experimental study to demonstrate its computational efficiency in comparison to existing attention methods. In particular,

we compare with a naive attention implementation from `PyTorch` (based on Vaswani et al. (2017)) as well as an IO-aware Flash-Attention (Dao et al., 2022; Dao, 2024). Our implementation is based on Triton.

Table 3: Time (in milliseconds) for different operations to compute for a fixed-sized batch of varying sequence length. Our method outperforms a naive attention implementation across all lengths while also outperforming Flash-Attention by a signficant margin as the sequence length increases.

| Method | Torch Attention | | Mamba | | Flash Attention | | ZETA | |
|---|---|---|---|---|---|---|---|---|
| Input Length | FWD | FWD+BWD | FWD | FWD+BWD | FWD | FWD+BWD | FWD | FWD+BWD |
| 4096 | 44.3 | 117.9 | 7.1 | 14.0 | 3.4 | 29.2 | 5.6 | 38.2 |
| 8192 | OOM | OOM | 11.8 | 23.0 | 12.8 | 111.5 | 11.0 | 76.4 |
| 16384 | OOM | OOM | 23.5 | 45.7 | 50.4 | 437.7 | 21.7 | 152.6 |
| 32768 | OOM | OOM | 47.3 | 91.8 | 198.2 | 1733.5 | 43.0 | 304.8 |
| 65536 | OOM | OOM | 94.0 | 183.7 | 805.3 | 7044.1 | 85.8 | 608.2 |

Table 4: Memory consumption (in MB) for different operations to compute for a fixed-sized batch of varying sequence length. Our method outperforms a naive attention implementation across all lengths while only marginally trailing a highly optimized Flash Attention implementation.

| Method | Torch Attention | | Mamba | | Flash Attention | | ZETA | |
|---|---|---|---|---|---|---|---|---|
| Input Length | FWD | FWD+BWD | FWD | FWD+BWD | FWD | FWD+BWD | FWD | FWD+BWD |
| 4096 | 17268.1 | 25972.1 | 574.2 | 632.2 | 886.1 | 1784.1 | 1314.1 | 1926.1 |
| 8192 | OOM | OOM | 904.2 | 1020.2 | 1528.1 | 3324.1 | 2382.1 | 3606.1 |
| 16384 | OOM | OOM | 1564.2 | 1776.2 | 2812.1 | 6404.1 | 4520.1 | 6968.2 |
| 32768 | OOM | OOM | 2884.2 | 3200.2 | 5380.1 | 12564.1 | 8796.2 | 13692.2 |
| 65536 | OOM | OOM | 5524.2 | 6048.2 | 10516.1 | 24884.1 | 17348.2 | 27140.3 |

Table 3 indicates the time required for both a forward pass as well as a forward-backward pass using our efficient ZETA implementation as well as the aforementioned attention implementations. We observe that our implementation significantly outspeeds a naive implementation of attention and do not suffer from out of memory issues while also outperforming Flash-Attention for long sequences, with a widening gap as the sequence length increases. This indicates both the computational efficiency of our method as well as serves as an empirical validation of the $\mathcal{O}(N \log N)$ complexity of ZETA which we previously justify theoretically. Furthermore, if we compare with Mamba (Gu & Dao, 2024), we demonstrate that ZETA has a faster forward pass while the forward-backward pass maintains a similar relative performance as the sequence length increases.

Table 4 meanwhile shows that ZETA uses less memory than a naive attention implementation while also only slightly utilizing more memory than a highly optimized Flash Attention implementation. Nevertheless, in comparison to a sequence model such as Mamba, all attention models predictably use more memory due to the use of softmax-type operations.

## 5 CONCLUSION

In this paper, we presented ZETA, a model designed to enhance the efficiency of top-$k$ attention by leveraging $Z$-order curves for parallel token selection in one-dimensional space, reducing both time and space complexity to $\mathcal{O}(N \log N)$. By carefully selecting the dimensionality of key and query pairs, ZETA effectively preserves relative distances, improving both locality and computational efficiency. Our comprehensive experiments on synthetic associative recall, LRA, and WIKITEXT-103 demonstrate that ZETA consistently matches or outperforms traditional attention mechanisms, making it particularly well-suited for long-sequence tasks that demand scalability and efficiency. Additionally, the introduction of the Adaptive Cauchy-Softmax mechanism enhances ZETA's flexibility, enabling it to handle long-range dependencies more effectively and efficiently. Overall, ZETA offers a robust, scalable, and efficient solution for sequence modeling, combining adaptive token selection with dynamic softmax to optimize performance across a range of tasks and datasets.

ACKNOWLEDGEMENTS

We appreciate constructive feedback from anonymous reviewers and meta-reviewers. This work is supported by the Natural Sciences and Engineering Research Council of Canada (NSERC), Discovery Grants program.

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

# A  THEORETICAL ANALYSIS

The recent paper, Reformer, proposed the Shared-$QK$ Transformer that shares the linear projection layer for keys and queries. It reduces the number of parameters and memory space while not affecting the model performance. https://arxiv.org/abs/2001.0445a very simple but efficient technique.

**Lemma A.1** (Johnson–Lindenstrauss Lemma). *For any $0 < \epsilon < 1$ and any integer $m$, let $d$ be a positive integer such that $d = \Omega(\frac{\ln m}{\epsilon^2})$. Then for any set $x$ of $m$ points in $\mathbb{R}^D$, there exists a map $f : \mathbb{R}^D \to \mathbb{R}^d$ such that for all $x_i, x_j \in \mathcal{X}$,*

$$(1 - \epsilon)\|x_i - x_j\|^2 \le \|f(x_i) - f(x_j)\|^2 \le (1 + \epsilon)\|x_i - x_j\|^2. \tag{8}$$

**Assumption A.2.** Let $\alpha \in \Delta_{m-1}$ be an element of the $m$-dimensional simplex, defined as $\Delta_{m-1} \triangleq \{\alpha \in \mathbb{R}^m \mid \alpha_i \ge 0, \sum_{i=1}^m \alpha_i = 1\}$. Assume that $h_{\text{attn}}$ equipped with $\alpha$ can achieve an optimal learnable similarity function $\Gamma$, where the attention scores are given by $\alpha = \text{softmax}(\Gamma(f_k(x_i), f_q(x)))$, such that $\Gamma$ is trained to be optimal to have the minimal expected risk: $\min_\alpha \|h_{\text{attn}}(x, S_x; \alpha) - y\|$, where $h_{\text{attn}}$ denotes the attention-based hypothesis, $x$ is the input, and $S_x$ is the context.

**Lemma A.3.** *Let $\alpha \in \mathbb{R}^m$, $\forall h_1 : \mathbb{R}^m \to \mathbb{R}$ and $\forall h_2 : \mathbb{R}^m \to \mathbb{R}$. Assume that $h_1(\alpha) \le h_2(\alpha)$. Then $\min_\alpha h_1(\alpha) \le \min_\alpha h_2(\alpha)$.*

*Proof.* Let $\alpha_1 = \arg\min_\alpha h_1(\alpha)$ and $\alpha_2 = \arg\min_\alpha h_2(\alpha)$. Then, $h_1(\alpha_2) \le h_2(\alpha_2)$ due to condition $h_1(\alpha) \le h_2(\alpha)$. We also have $h_1(\alpha_1) \le h_1(\alpha_2)$ due to condition $\alpha_1$ achieving the minimum of $h_1$. To sum up, we have $h_1(\alpha_1) \le h_2(\alpha_2)$. $\qquad\square$

**Lemma A.4.** *(Shalev-Shwartz & Ben-David, 2014) Let $C_1, \ldots, C_r$ be a collection of subsets of some domain set, $\mathcal{X}$. Let $S$ be a sequence of $m$ points sampled i.i.d. according to some probability distribution, $\mathcal{D}$, over $\mathcal{X}$. Then,*

$$\mathbb{E}_{S \sim \mathcal{D}^m} \left[ \sum_{i:C_i \cap S = \emptyset} \mathbb{P}[C_i] \right] \le \frac{r}{me}.$$

*Proof.* From the linearity of expectation, we can rewrite

$$\mathbb{E}_S \left[ \sum_{i:C_i \cap S = \emptyset} \mathbb{P}[C_i] \right] = \sum_{i=1}^r \mathbb{P}[C_i]\mathbb{E}_S \left[ \mathbf{1}_{C_i \cap S = \emptyset} \right].$$

Next, for each $i$ we have

$$\mathbb{E}_S \left[ \mathbf{1}_{C_i \cap S = \emptyset} \right] = \mathbb{P}_S[C_i \cap S = \emptyset] = (1 - \mathbb{P}[C_i])^m \le e^{-\mathbb{P}[C_i]m}.$$

Combining the preceding two equations, we get

$$\mathbb{E}_S \left[ \sum_{i:C_i \cap S = \emptyset} \mathbb{P}[C_i] \right] \le \sum_{i=1}^r \mathbb{P}[C_i]e^{-\mathbb{P}[C_i]m} \le r \max_i \mathbb{P}[C_i]e^{-\mathbb{P}[C_i]m}.$$

Finally, by elementary calculus, $\max_a ae^{-ma} \le \frac{1}{me}$, concluding the proof. $\qquad\square$

**Theorem A.5.** *Let $\mathcal{X} \in \mathbb{R}^d$, $\mathcal{Y} \in \mathbb{R}^D$, and $\mathcal{D}$ be a distribution over $\mathcal{X} \times \mathcal{Y}$ for which the conditional probability function, $h : \mathbb{R}^d \to \mathbb{R}^D$, is a $l$-Lipschitz function. Let $h$ denote a hypothesis, and $h_{\text{attn}}$ denote the one-layer attention model to aggregate the predictions of a sample set $S \sim \mathcal{D}^m$ to predict for another i.i.d sample $x$. Specifically, here we assume the same linear map for a key mapping $f_k$ and a query mapping $f_q$ as $f : \mathbb{R}^d \to [-B, B]^{d_K}$ where $B$ is the bound of projection of $x$ by $f$, and we assume the value mapping $f_v$ be the Bayes optimal rule $h^* = \mathbb{E}[Y|X = x]$, which is the*

*hypothesis that minimizes $L_\mathcal{D}(h)$ over all functions. Then, the expected risk of the regression tasks $L_\mathcal{D}(h) = \mathbb{E}_{(x,y)\sim\mathcal{D}}\|h(x) - y\|$ with $h$ as $h_{\mathrm{attn}}$ can be upper bounded by*

$$\mathbb{E}_{S\sim\mathcal{D}^m}\left[L_\mathcal{D}(h_{\mathrm{attn}})\right] \le L_\mathcal{D}(h^*) + \frac{4lc\sqrt{d_K}Bm^{-1/(d_K+1)}}{\sqrt{1 - \sqrt{\frac{C\ln m}{d_K}}}}.$$

*Proof.* $\mathbb{E}_S[L_\mathcal{D}(h)]$ is the root mean square error (RMSE) between the prediction and $y$ conditioned on a sampled set $S$ and an additional example $(x,y)$, such that the label of $\pi_1(x)$ is different from $y$. In other words, we first sample $m$ examples, $S_x = \{x_1, \ldots, x_m\}$, according to $\mathcal{D}_\mathcal{X}$, and an additional example, $(x, y)$. It follows that

$$\mathbb{E}_S[L_\mathcal{D}(h_{\mathrm{attn}})] = \mathbb{E}_{S_x\sim\mathcal{D}_\mathcal{X}^m, x\sim\mathcal{D}_\mathcal{X}}\mathbb{E}_{y\sim\mathcal{D}_{\mathcal{Y}|\mathcal{X}}}\min_\alpha \|h_{\mathrm{attn}}(x, S_x; \alpha) - y\|$$

$$= \mathbb{E}_{S_x\sim\mathcal{D}_\mathcal{X}^m, x\sim\mathcal{D}_\mathcal{X}}\mathbb{E}_{y\sim\mathcal{D}_{\mathcal{Y}|\mathcal{X}}}\min_\alpha \|\sum_{i=1}^m \alpha_i h^*(x_i) - y\|$$

$$\le \mathbb{E}_{S_x\sim\mathcal{D}_\mathcal{X}^m, x\sim\mathcal{D}_\mathcal{X}}\mathbb{E}_{y\sim\mathcal{D}_{\mathcal{Y}|\mathcal{X}}}\left[\min_\alpha \|\sum_{i=1}^m \alpha_i h^*(x_i) - h^*(x)\| + \|h^*(x) - y\|\right]$$

$$= L_\mathcal{D}(h^*) + \mathbb{E}_{S_x\sim\mathcal{D}_\mathcal{X}^m, x\sim\mathcal{D}_\mathcal{X}}\mathbb{E}_{y\sim\mathcal{D}_{\mathcal{Y}|\mathcal{X}}}\min_\alpha \|\sum_{i=1}^m \alpha_i h^*(x_i) - h^*(x)\|,$$

where $\alpha_i$ is the attention score between $x_i$ and $x$. The inequality follows the Cauchy-Schwarz inequality and Lemma A.3.

$$\mathbb{E}_{S_x\sim\mathcal{D}_\mathcal{X}^m, x\sim\mathcal{D}_\mathcal{X}}\mathbb{E}_{y\sim\mathcal{D}_{\mathcal{Y}|\mathcal{X}}}\min_\alpha \|\sum_{i=1}^m \alpha_i h^*(x_i) - h^*(x)\| \le \mathbb{E}_{S_x\sim\mathcal{D}_\mathcal{X}^m, x\sim\mathcal{D}_\mathcal{X}}\mathbb{E}_{y\sim\mathcal{D}_{\mathcal{Y}|\mathcal{X}}}\min_\alpha \sum_{i=1}^m \alpha_i\|h^*(x_i) - h^*(x)\|$$

$$\le \mathbb{E}_{S_x\sim\mathcal{D}_\mathcal{X}^m, x\sim\mathcal{D}_\mathcal{X}}\mathbb{E}_{y\sim\mathcal{D}_{\mathcal{Y}|\mathcal{X}}}\left[l\cdot\min_\alpha \sum_{i=1}^m \alpha_i\|x_i - x\|\right] \le \mathbb{E}_{S_x\sim\mathcal{D}_\mathcal{X}^m, x\sim\mathcal{D}_\mathcal{X}}\mathbb{E}_{y\sim\mathcal{D}_{\mathcal{Y}|\mathcal{X}}}\left[l\cdot\min_\alpha \sum_{i=1}^m \alpha_i\frac{\|f(x_i) - f(x)\|}{\sqrt{1 - \sqrt{\frac{C\ln m}{d_K}}}}\right]$$

$$= \mathbb{E}_{S_x\sim\mathcal{D}_\mathcal{X}^m, x\sim\mathcal{D}_\mathcal{X}}\mathbb{E}_{y\sim\mathcal{D}_{\mathcal{Y}|\mathcal{X}}}\left[l\cdot\min_\alpha \sum_{i=1}^m \alpha_i\frac{\|k_i - q\|}{\sqrt{1 - \sqrt{\frac{C\ln m}{d_K}}}}\right]$$

The first inequality follows from the Cauchy-Schwarz inequality; the second inequality follows from the $l$-Lipschitzness of $h^*$; the third inequality follows from the Johnson–Lindenstrauss Lemma, where we define $\epsilon$ in the Johnson–Lindenstrauss Lemma as $\epsilon = \sqrt{\frac{C\ln m}{d_K}}$, where $C$ is a constant; $k_i = f(x_i)$ and $q = f(x)$. It is obvious that

$$\min_\alpha \sum_{i=1}^m \alpha_i\|k_i - q\| = \|k_{\pi_1(q)} - q\|$$

where $k_{\pi_1(q)}$ is the closest $k_i$ to $q$. Thus we have

$$\mathbb{E}_{S_x\sim\mathcal{D}_\mathcal{X}^m, x\sim\mathcal{D}_\mathcal{X}}\mathbb{E}_{y\sim\mathcal{D}_{\mathcal{Y}|\mathcal{X}}}\min_\alpha \left\|\sum_{i=1}^m \alpha_i h^*(x_i) - h^*(x)\right\| = \mathbb{E}_{S_x\sim\mathcal{D}_\mathcal{X}^m, x\sim\mathcal{D}_\mathcal{X}}\mathbb{E}_{y\sim\mathcal{D}_{\mathcal{Y}|\mathcal{X}}}\left[l\cdot\frac{\|k_{\pi_1(q)} - q\|}{\sqrt{1 - \sqrt{\frac{C\ln m}{d_K}}}}.\right]$$

Fix some $\zeta = 2B/T$. For some integer $T$, let $r = T^{d_K}$ and $C_1, \ldots, C_r$ be the cover of the set $\mathcal{X}$ using boxes of length $\zeta$. Namely, for every $(\xi_1, \ldots, \xi_d) \in [T]^d$, there exists a set $C_i$ of the form $\{k : \forall j, k_j \in [2B(\xi_j - 1)/T, 2B\xi_j/T]\}$.

For each $k, q$ in the same box we have $\|k - q\| \le \sqrt{d_K}B\zeta$. Otherwise, $\|k - q\| \le \sqrt{d_K}B$. Therefore,

$$\mathbb{E}_{S_x,x}\left[\|k_{\pi_1(q)} - q\|\right] \leq \mathbb{E}_S\left[\mathbb{P}\left[\bigcup_{i:C_i \cap S_x=\emptyset} C_i\right]\sqrt{d_K}B + \mathbb{P}\left[\bigcup_{i:C_i \cap S_x \neq \emptyset} C_i\right]B\zeta\sqrt{d_K}\right],$$

and by combining Lemma A.4 with the trivial bound

$$\mathbb{P}[\bigcup_{i:C_i \cap S_x \neq \emptyset} C_i] \leq 1,$$

we get that

$$\mathbb{E}_{\mathbf{x},S_x}\left[\|k_{\pi_1(q)} - q\|\right] \leq \sqrt{d_K}B\left(\frac{r}{me} + \zeta\right).$$

Since the number of boxes is $r = (1/\zeta)^{d_K}$, it follows that

$$\mathbb{E}_{S_x,\mathbf{x}}\left[\|k_{\pi_1(q)} - q\|\right] \leq \sqrt{d_K}B\left(\frac{2^{d_K}\zeta^{-d_K}}{me} + \zeta\right).$$

Setting $\zeta = 2m^{-1/(d_K+1)}$ and noting that

$$\frac{2^{d_K}\zeta^{-d}}{me} + \zeta = \frac{2^{d_K}2^{-d_K}m^{d_K/(d_K+1)}}{me} + 2m^{-1/(d_K+1)}$$

$$= m^{-1/(d_K+1)}\left(\frac{1}{e} + 2\right) \leq 4m^{-1/(d_K+1)},$$

then combining the preceding with previous results, we obtain

$$\mathbb{E}_S\left[L_\mathcal{D}(h_{\text{attn}})\right] \leq L_\mathcal{D}(h^\star) + \frac{4lc\sqrt{d_K}Bm^{-1/(d_K+1)}}{\sqrt{1 - \sqrt{\frac{C\ln m}{d_K}}}}.$$

$\square$

# B  AN ILLUSTRATIVE EXAMPLE OF EFFICIENT TOP-$k$ SELECTION VIA $Z$-ORDER CURVE CHUNKING

In ZETA, the chunking process is a crucial step for efficient key-query matching in large-scale attention mechanisms. The process begins by projecting the high-dimensional keys into a one-dimensional space using $Z$-order curves, which preserve spatial locality. These projections create a linear representation of the keys, as shown in the second row of the figure.

After projection, the keys are radix sorted in $\mathcal{O}(N)$ into ascending order of their $Z$-order integer values, enabling efficient binary searching in $\mathcal{O}(N)$ for the most relevant keys in the one-dimensional space. Sorting ensures that keys that are spatially close in the original multi-dimensional space remain close in the projected space, making the retrieval process computationally efficient.

Next, the sorted keys are partitioned into chunks. Each chunk contains a fixed number of keys, and queries are matched with the keys within their respective chunks. This chunk-based structure facilitates parallel processing, where each query can efficiently search for its top-$k$ nearest keys within a local subset of the sorted key space. For instance, in the figure, the 5th query retrieves its top-4 keys from the chunks containing its most relevant keys.

This chunking approach significantly reduces the computational overhead compared to searching the entire key space for each query, while leveraging the locality-preserving properties of $Z$-order curves. The result is a scalable and efficient mechanism for top-$k$ selection in ZETA, ensuring both speed and accuracy for attention-based tasks.

The pseudo-code in Algorithm 1 outlines the ZETA Top-$k$ Attention mechanism, which combines $Z$-order curve projections with chunk-based sorting to efficiently identify and retrieve the top-$k$ nearest neighbors while maintaining causal constraints. It provides a structured approach to reduce computational overhead by limiting the search space to relevant chunks, ensuring both efficiency and adherence to sequence masking requirements.

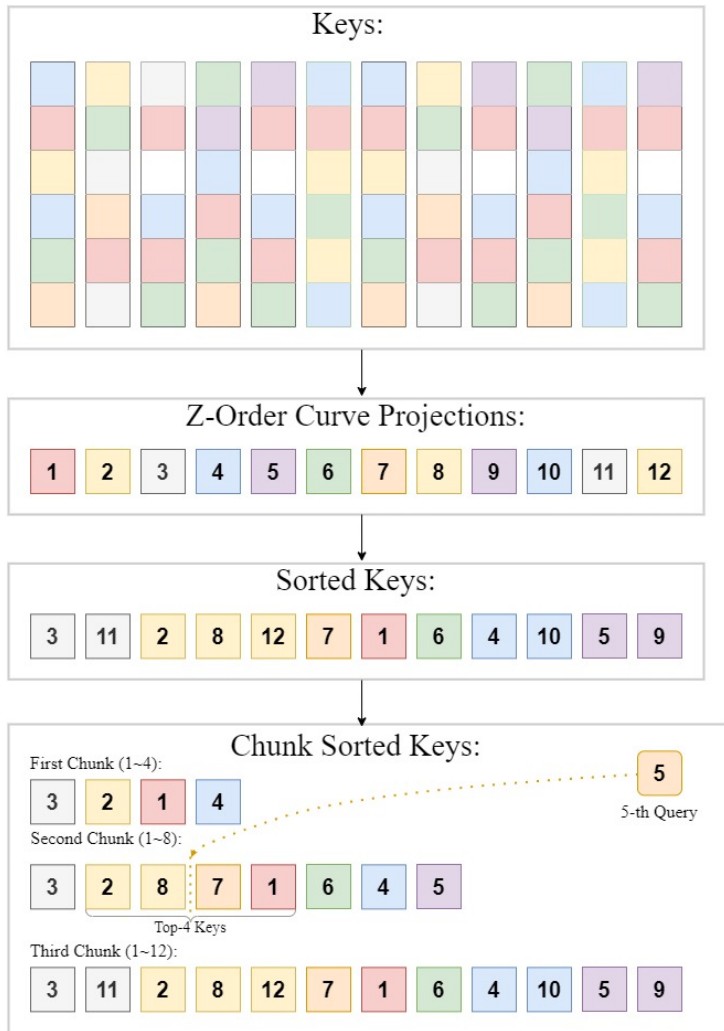

Figure 4: Illustration of the chunking process in ZETA: Keys are projected into one-dimensional space using $Z$-order curves, sorted, and partitioned into chunks for efficient retrieval of top-$k$ keys for each query.

## C EXPERIMENTS DETAILS

The ZETA model configuration generally involves setting the number of chunks to values such as $4, 8, 16, 32$ depending on the sequence length, which provides a flexible way to handle different input scales effectively. This chunking strategy facilitates parallelism in processing, allowing for efficient memory use and computational speedup during attention operations. The hidden dimension, $d_V$, is typically set to 256 or 512 with 8 attention heads when working with LRA datasets. However, for larger and more complex datasets such as WIKITEXT-103, the hidden dimension is increased to $d_V = 768$ with 12 attention heads to ensure that the model has sufficient capacity to learn intricate long-range dependencies effectively. Additionally, the dimensions of keys and queries are kept significantly lower at $d_K = d_Q = 3$, which aids in reducing the computational burden and mitigates the "curse of dimensionality" while still preserving enough information for efficient attention computation. This choice of dimensions strikes a balance between model efficiency and effectiveness, making ZETA well-suited for long-sequence modeling tasks.

---

**Algorithm 1:** ZETA Top-$k$ Attention Using $Z$-order Curves and Chunking

---

**Data:** Keys $\mathcal{K}$ of size $M \times d$, sequence length $N$, chunk size $M$, query index $i$, top-$k$ value $k$
**Result:** Top-$k$ nearest keys for each query

**Step 1: $Z$-order Curve Projection**;
**for** *each key vector in $\mathcal{K}$* **do**
   | Project each $d$-dimensional key vector into a one-dimensional representation using $Z$-order
   |   curve projection;
**end**
**Step 2: Sorting of Keys**;
Sort the projected one-dimensional keys in ascending order;
**Step 3: Chunk Division**;
Divide the sorted keys into multiple chunks based on the chunk size $M$;
**Step 4: Chunk-wise Causal Masking**;
**for** *each query at position $i$* **do**
   | Determine chunk index $m = \lfloor \frac{i}{M} \rfloor$;
   | Search only in the first $m$ chunks for the top-$k$ keys to attend to;
   | Exclude keys from positions $j > m \times M$;
**end**
**Step 5: Nearest Neighbor Search**;
**for** *each query* **do**
   | Perform nearest neighbor search within the selected chunks to find top-$k$ tokens;
   | Collect the indices of the nearest neighbors $I_q$ while maintaining causal constraints;
**end**
**return** *Top-k nearest keys for each query*

---

# D    I/O-AWARE ZETA OPTIMIZED WITH TRITON

Our Triton implementation of ZETA focuses on improving the efficiency of sparse attention through customized kernel programming. We leverage Triton to create specialized GPU kernels for top-$k$ sparse attention. The `sparse_topk_attention_kernel` and its corresponding backward pass kernel `sparse_topk_attention_backward_kernel` are implemented using the Triton JIT (Just-In-Time) compiler. This approach allows for significant speedup by optimizing memory access patterns and reducing I/O overhead during the computation. The kernel is tuned to different configurations, like "block_size" and "num_warps", which directly influence how GPU resources are allocated. Especially, we compute the mean vector of history tokens in the block of the current kernel, instead of computing global mean vectors, which effectively reduce overheads. he `@triton.autotune` decorator is used to evaluate multiple kernel configurations for optimal performance, making sure that GPU resources are well-utilized depending on sequence length and other parameters.

One key challenge addressed is efficient indexing for large tensors in the backward and forward computations. In Triton, indexing is achieved via program IDs, `tl.program_id()`, that are used to identify which part of the workload is being computed by each block or thread, ensuring that parallelism is effectively exploited. The Triton kernel computes Cauchy Softmax for top-$k$ KV pairs for each query, employing a specialized kernel to access only the most relevant $k$ keys during the attention process. This reduces the computational complexity compared to a full attention mechanism, and Triton's low-level bit manipulation operations (`tl.load` and `tl.store`) are used for fast data retrieval.

The sparse Attention Mechanism is computed by considering only the top-$k$ keys per query, which significantly reduces the computational load. The `sparse_topk_attention_kernel` involves efficient gathering of keys and values based on top-$k$ indices. The indices are pre-computed in a sorted order, which enables efficient retrieval without scanning the entire key space.

The Triton kernel also includes custom backward functions to handle the gradient flow. The backward kernel `sparse_topk_attention_backward_kernel` computes gradients for each parameter involved in the sparse attention operation, including $q, k, v$ and the learnable parameter $\gamma$. Triton's

tl.atomic_add is used to accumulate gradients, ensuring that all updates to shared memory locations are synchronized.

During the forward pass, intermediate values such as the Euclidean distances and normalization constants are stored. These values are reused in the backward pass, which reduces redundant computations and accelerates the training process.

By using Triton, we managed to reduce the I/O overhead that traditional PyTorch operations faced, especially during backward computations. Furthermore, we used fused kernels to mitigate the overhead associated with multiple indexing operations. This fusion helps in reducing the number of kernel launches, which translates to reduced latency and faster execution, as Triton allows more control over memory coalescing and efficient block-wise operations.

Overall, the Triton-based implementation in ZETA allows for a more scalable sparse attention mechanism that retains the benefits of locality preservation through $Z$-order Curves while significantly reducing computational and I/O bottlenecks. This makes the ZETA attention more suitable for long-sequence tasks where traditional transformers are too resource-intensive.

## E  GRADIENT DERIVATION FOR THE BACKWARD PASS OF TRITON ZETA

### E.1  UPDATED ATTENTION MECHANISM WITH SPARSE ATTENTION

We introduce sparse attention by computing values and keys from an index set $I_i$ of top-$k$ tokens specific to each query $\mathbf{q}_i$. The unnormalized attention scores are defined as:

$$S_{ij} = \frac{1}{D_{ij} + \varepsilon}, \quad \text{for } j \in I_i \tag{9}$$

where:

- $D_{ij} = \|\mathbf{q}_i - \mathbf{k}_j\|^2$
- $\varepsilon = \gamma^2$ is a trainable scalar parameter

Define:

$$\delta_{ij} = D_{ij} + \varepsilon \tag{10}$$

The steps of the attention mechanism are:

1. **Compute Pairwise Distances**:
$$D_{ij} = \|\mathbf{q}_i - \mathbf{k}_j\|^2, \quad \text{for } j \in I_i \tag{11}$$

2. **Compute Unnormalized Attention Scores**:
$$S_{ij} = \frac{1}{\delta_{ij}}, \quad \text{for } j \in I_i \tag{12}$$

3. **Normalize Attention Weights**:
$$Z_i = \sum_{j \in I_i} S_{ij} \tag{13}$$
$$A_{ij} = \frac{S_{ij}}{Z_i}, \quad \text{for } j \in I_i \tag{14}$$

4. **Compute Output**:
$$\mathbf{o}_i = \sum_{j \in I_i} A_{ij} \mathbf{v}_j \tag{15}$$

Our goal is to compute the gradients of the output $\mathbf{o}_i$ with respect to $\mathbf{q}_i$, $\mathbf{k}_j$, $\mathbf{v}_j$, and $\varepsilon$, considering the sparse attention.

### E.2 GRADIENTS WITH RESPECT TO $\mathbf{Q}$, $\mathbf{K}$, $\mathbf{V}$, AND $\varepsilon$

The gradient of the output $\mathbf{o}_i$ with respect to $\mathbf{v}_j$ is:

$$\frac{\partial \mathbf{o}_i}{\partial \mathbf{v}_j} = \begin{cases} A_{ij}, & \text{if } j \in I_i \\ 0, & \text{otherwise} \end{cases} \tag{16}$$

**Derivative of $D_{ij}$ with respect to $\mathbf{q}_i$ and $\mathbf{k}_j$:**

$$\frac{\partial D_{ij}}{\partial \mathbf{q}_i} = 2(\mathbf{q}_i - \mathbf{k}_j), \quad \text{for } j \in I_i \tag{17}$$

$$\frac{\partial D_{ij}}{\partial \mathbf{k}_j} = \begin{cases} -2(\mathbf{q}_i - \mathbf{k}_j), & \text{if } j \in I_i \\ 0, & \text{otherwise} \end{cases} \tag{18}$$

**Derivative of $S_{ij}$:**

First, compute the derivative of $S_{ij}$ with respect to $\delta_{ij}$:

$$\frac{\partial S_{ij}}{\partial \delta_{ij}} = -\frac{1}{\delta_{ij}^2}, \quad \text{for } j \in I_i \tag{19}$$

Compute the derivative of $\delta_{ij}$ with respect to $D_{ij}$ and $\varepsilon$:

$$\frac{\partial \delta_{ij}}{\partial D_{ij}} = 1, \quad \text{for } j \in I_i \tag{20}$$

$$\frac{\partial \delta_{ij}}{\partial \varepsilon} = 1, \quad \text{for } j \in I_i \tag{21}$$

Now, compute the derivative of $S_{ij}$ with respect to $D_{ij}$ and $\varepsilon$:

$$\frac{\partial S_{ij}}{\partial D_{ij}} = -\frac{1}{\delta_{ij}^2}, \quad \text{for } j \in I_i \tag{22}$$

$$\frac{\partial S_{ij}}{\partial \varepsilon} = -\frac{1}{\delta_{ij}^2}, \quad \text{for } j \in I_i \tag{23}$$

**Derivative of $S_{ij}$ with respect to $\mathbf{q}_i$ and $\mathbf{k}_j$:**

$$\frac{\partial S_{ij}}{\partial \mathbf{q}_i} = -\frac{2(\mathbf{q}_i - \mathbf{k}_j)}{\delta_{ij}^2}, \quad \text{for } j \in I_i \tag{24}$$

$$\frac{\partial S_{ij}}{\partial \mathbf{k}_j} = \begin{cases} \frac{2(\mathbf{q}_i - \mathbf{k}_j)}{\delta_{ij}^2}, & \text{if } j \in I_i \\ 0, & \text{otherwise} \end{cases} \tag{25}$$

**Derivative of $Z_i$ with respect to $S_{ij}$:**

$$\frac{\partial Z_i}{\partial S_{ij}} = 1, \quad \text{for } j \in I_i \tag{26}$$

**Derivative of $A_{ij}$ with respect to $S_{ij}$:**

$$\frac{\partial A_{ij}}{\partial S_{ij}} = \frac{Z_i - S_{ij}}{Z_i^2}, \quad \text{for } j \in I_i \tag{27}$$

**Derivative of $A_{ij}$ with respect to $S_{il}$ for $l \neq j$:**

$$\frac{\partial A_{ij}}{\partial S_{il}} = \begin{cases} -\dfrac{S_{ij}}{Z_i^2}, & \text{if } l \in I_i \\ 0, & \text{otherwise} \end{cases} \tag{28}$$

**Gradient of $\mathbf{o}_i$ with respect to $S_{il}$:**

$$\frac{\partial \mathbf{o}_i}{\partial S_{il}} = \sum_{j \in I_i} \mathbf{v}_j \frac{\partial A_{ij}}{\partial S_{il}} \tag{29}$$

$$= \frac{\mathbf{v}_l - \mathbf{o}_i}{Z_i}, \quad \text{for } l \in I_i \tag{30}$$

**Gradient with Respect to $\mathbf{q}_i$:**

$$\frac{\partial \mathbf{o}_i}{\partial \mathbf{q}_i} = \sum_{l \in I_i} \frac{\partial \mathbf{o}_i}{\partial S_{il}} \cdot \frac{\partial S_{il}}{\partial \mathbf{q}_i} \tag{31}$$

$$= -2 \sum_{l \in I_i} \frac{\mathbf{v}_l - \mathbf{o}_i}{Z_i} \cdot \frac{(\mathbf{q}_i - \mathbf{k}_l)}{\delta_{il}^2} \tag{32}$$

**Gradient with Respect to $\mathbf{k}_j$:**

$$\frac{\partial \mathbf{o}_i}{\partial \mathbf{k}_j} = \begin{cases} 2\dfrac{\mathbf{v}_j - \mathbf{o}_i}{Z_i} \cdot \dfrac{(\mathbf{q}_i - \mathbf{k}_j)}{\delta_{ij}^2}, & \text{if } j \in I_i \\ 0, & \text{otherwise} \end{cases} \tag{33}$$

The total gradient with respect to $\mathbf{k}_j$ is:

$$\frac{\partial L}{\partial \mathbf{k}_j} = \sum_{i:j \in I_i} \left( 2\frac{\partial L}{\partial \mathbf{o}_i} \cdot \frac{\mathbf{v}_j - \mathbf{o}_i}{Z_i} \cdot \frac{(\mathbf{q}_i - \mathbf{k}_j)}{\delta_{ij}^2} \right) \tag{34}$$

**Gradient with Respect to $\varepsilon$:**

$$\frac{\partial \mathbf{o}_i}{\partial \varepsilon} = \sum_{l \in I_i} \frac{\partial \mathbf{o}_i}{\partial S_{il}} \cdot \frac{\partial S_{il}}{\partial \varepsilon} \tag{35}$$

$$= -\sum_{l \in I_i} \frac{\mathbf{v}_l - \mathbf{o}_i}{Z_i} \cdot \frac{1}{\delta_{il}^2} \tag{36}$$

The total gradient with respect to $\varepsilon$ is:

$$\frac{\partial L}{\partial \varepsilon} = -\sum_i \left( \frac{\partial L}{\partial \mathbf{o}_i} \sum_{l \in I_i} \frac{\mathbf{v}_l - \mathbf{o}_i}{Z_i} \cdot \frac{1}{\delta_{il}^2} \right) \tag{37}$$

### E.3 SUMMARY OF GRADIENT COMPUTATIONS

1. **Compute $D_{ij}$, $\delta_{ij}$, $S_{ij}$, $Z_i$, $A_{ij}$, and $\mathbf{o}_i$:**

$$D_{ij} = \|\mathbf{q}_i - \mathbf{k}_j\|^2, \quad \text{for } j \in I_i \tag{38}$$

$$\delta_{ij} = D_{ij} + \varepsilon, \quad \text{for } j \in I_i \tag{39}$$

$$S_{ij} = \frac{1}{\delta_{ij}}, \quad \text{for } j \in I_i \tag{40}$$

$$Z_i = \sum_{j \in I_i} S_{ij} \tag{41}$$

$$A_{ij} = \frac{S_{ij}}{Z_i}, \quad \text{for } j \in I_i \tag{42}$$

$$\mathbf{o}_i = \sum_{j \in I_i} A_{ij} \mathbf{v}_j \tag{43}$$

2. **Compute Gradient with Respect to V:**

$$\frac{\partial L}{\partial \mathbf{v}_j} = \sum_{i:j \in I_i} \frac{\partial L}{\partial \mathbf{o}_i} \cdot A_{ij} \tag{44}$$

3. **Compute Gradient with Respect to Q:**

$$\frac{\partial L}{\partial \mathbf{q}_i} = -2 \left( \frac{\partial L}{\partial \mathbf{o}_i} \right) \sum_{j \in I_i} \frac{\mathbf{v}_j - \mathbf{o}_i}{Z_i} \cdot \frac{(\mathbf{q}_i - \mathbf{k}_j)}{\delta_{ij}^2} \tag{45}$$

4. **Compute Gradient with Respect to K:**

$$\frac{\partial L}{\partial \mathbf{k}_j} = \sum_{i:j \in I_i} \left( 2 \frac{\partial L}{\partial \mathbf{o}_i} \cdot \frac{\mathbf{v}_j - \mathbf{o}_i}{Z_i} \cdot \frac{(\mathbf{q}_i - \mathbf{k}_j)}{\delta_{ij}^2} \right) \tag{46}$$

5. **Compute Gradient with Respect to $\varepsilon$:**

$$\frac{\partial L}{\partial \varepsilon} = -\sum_i \left( \frac{\partial L}{\partial \mathbf{o}_i} \sum_{j \in I_i} \frac{\mathbf{v}_j - \mathbf{o}_i}{Z_i} \cdot \frac{1}{\delta_{ij}^2} \right) \tag{47}$$

### E.4 IMPLEMENTATION NOTES

- **Sparse Attention Index Set $I_i$:**
  - $I_i$ is the set of indices that query $\mathbf{q}_i$ attends to, which is collected from $Z$-order Curve projected sequences.
  - The attention computations and gradient updates are performed only over $j \in I_i$.
- **Numerical Stability:**
  - The addition of $\varepsilon$ ensures that $\delta_{ij} > 0$ if $\varepsilon > 0$, preventing division by zero.
  - Ensure that $\varepsilon$ remains positive during training, possibly by parameterizing $\varepsilon = \exp(\theta)$ where $\theta$ is unconstrained.
- **Efficient Computation:**
  - Utilize sparse matrix representations to handle the index sets $I_i$ efficiently.
  - Use vectorized operations and appropriate masking to perform computations only over the valid indices.

## F LIMITATIONS

Given that our method is a top-$k$ attention mechanism, there are some shared limitations between our method and that of prior work that deals with attention, such as there still potentially being higher chances to ignore attention to important information (with low attention scores) than full attention methods given the use of only the top-$k$ tokens.

# G    ADDITIONAL EXPERIMENTS

## G.1    ABLATION ON ATTENTION PERFORMANCE WITH VARYING $d_K$ ON LRA

We expand on the ablation study presented in Figure 2(b), focusing on the ListOps and Image tasks from the Long Range Arena (LRA) benchmark. Specifically, we examine the impact of varying the dimensionality of the keys and queries ($d_K$) on attention performance. The results are summarized in Table 5.

Our experimental findings indicate that the performance remains relatively consistent for $d_K \geq 3$, whereas a noticeable decline is observed for $d_K < 3$. This supports our hypothesis that, unlike value vectors, the keys and queries predominantly encode positional information rather than intricate semantic features. As such, reducing $d_K$ to a small value allows us to effectively lower computational costs without incurring a significant loss in performance. This insight guided our selection of $d_K$ values in subsequent experiments.

| $d_K$ | 1 | 2 | 3 | 32 | 64 | 128 |
|---|---|---|---|---|---|---|
| ListOps | 31.04 | 34.79 | 36.06 | 36.19 | 36.32 | 36.37 |
| Image | 37.6 | 40.23 | 42.64 | 42.72 | 42.51 | 42.44 |

Table 5: Performance metrics for different values of $d_K$.

## G.2    PERFORMANCE OF ZETA USING DIFFERENT SIMILARITY METRIC

We utilize Euclidean distance for k-nearest neighbor (k-NN) searches to identify the top-$k$ attended tokens, as it is particularly well-suited for this purpose. In contrast, dot-product similarity cannot be directly employed for k-NN searches without normalization, as highlighted in (Mao et al., 2024). Our experimental results also indicate that normalized dot-product similarity performs worse than Euclidean distance, as demonstrated in additional MQAR experiments below.

Specifically, Euclidean distance-based methods, such as Negative Euclidean with traditional softmax and Cauchy Softmax, consistently outperform dot-product-based methods for top-$k$ attention when using a small dimensionality ($d_k \leq 4$), which is the setting adopted in ZETA.

| $d_k$ | 1 | 2 | 3 | 4 |
|---|---|---|---|---|
| Negative Euclidean | 64.6 | 99.4 | 99.4 | 99.3 |
| Inverse Euclidean | 22.9 | 81.3 | 99.6 | 99.9 |
| Cauchy Softmax | 74.5 | 99.6 | 99.5 | 99.2 |
| Normalized Dot Prod | 62.6 | 92.6 | 99.3 | 99.1 |

Table 6: Performance using various different similarity metrics

