# OpenReview forum: "ZETA: Leveraging $Z$-order Curves for Efficient Top-$k$ Attention"
_ICLR.cc/2025/Conference — ICLR 2025 Poster_

### Official Review · Reviewer_xJi4 · 2024-10-18

**Soundness:** 4
**Presentation:** 3
**Contribution:** 4
**Rating:** 8
**Confidence:** 4

**Summary:**

This paper introduced ZETA, a model to enhance the top-k attention using Z-order curves, reducing the complexity from N^2 to N log N. ZETA's query & key dimensions are much smaller than the value dimension. This is to ensure the speed and locality preservation of the Z-order curves. The authors also replaced the Softmax operation with the Cauchy-Softmax to enhance the performance of ZETA. Experiments showed that ZETA matched the performance of the attention models on Associative Recall while outperforming the attention models on Long-Range Arena and WikiText-103 language modeling.

**Strengths:**

* The paper is well-written and the method is believable.
* Compared with the attention models, the ZETA model matches the performance on Associative Recall while outperforming on Long-Range Arena and WikiText-103 language modeling.

**Weaknesses:**

* The effect of $d_K$ needs clarification.
  - In section 3.2.1, line 202, the authors mentioned: "a lower $d_K$ loses locality between tokens, which is crucial for efficient query".
  - However, in section 4.4, line 454, the authors mentioned that "Lower $d_K$ values exhibit a higher level of locality preservation across all sample sizes."
  - There seems to be a contradiction between these two statements, so the effect of $d_K$ needs clarification.
* Can the authors clarify the binary expansion? What is the precision of the original d-dimensional vector?
  - Eq. (4) introduced Z-order curves, which rearrange a d-dimensional vector into a one-dimensional binary representation.
  - Suppose d=3. Suppose the precision of the numbers in the d-dimensional vector is 32 bits (e.g., the float32). Then the resulting Z-order curves will have 3*32=96 bits.
  - Will this result in an overflow? If not, how did you set the precision of the original d-dimensional vector?
* Can the authors provide a comparison of the time and space usages for ZETA model and the attention-based models?
  - The authors claimed that ZETA reduces the time and space complexity to N log N. So I assume that ZETA runs faster and uses less memory. However, there is no actual number on this. So, the run-time metrics on time and space are needed.

**Questions:**

* The recent state-of-the-art LLMs have at least 100k context length. Gemini 1.5 Pro even supports 2M context length. How does this impact the selection of your $d_K$ and other hyperparameters? Will $d_K$ go to 2 or even 1?

---

> ### Author Response · Authors · 2024-11-22
> **Response to Reviewer xJi4 (1/3)**
>
> We thank the reviewer for their work in providing an assessment of our work. We are very appreciative that they find our manuscript well written and our results convincing and believable. We hope our response below can better resolve remaining misunderstandings and clarify any points that remain potentially unclear to the reviewer.
>
> -----
>
> > W1. The effect of $d_K$ needs further clarification.
>
> A1. **Effect of $d_K$** We are happy to clarify this potential misunderstanding, which may stem from wording that could be improved. In Line 202, we state that $d_K$ should not be too small, such as 1, as this would compromise locality. In Line 454, we discuss the ablation study on Z-Order Curves, highlighting that Z-Order Curves can better preserve locality with a lower $d_K$.
>
> To summarize, the process involves two steps:
>
> 1. Projecting inputs into keys: This requires $d_K \geq 3$ to preserve locality effectively.
> 2. Mapping keys into a 1-dimensional space using Z-Order Curves: This step benefits from a lower $d_K$ to avoid significant locality loss in the 1-dimensional projection.
>
> We argue that $d_K$ can be set to a much smaller dimension than $d_V$. However, an extremely small $d_K$, such as $d_K = 1$, deteriorates performance. As shown in Figure 2(b), performance drops when $d_K < 3$ but remains relatively stable for $d_K \geq 3$.
>
> This observation highlights a trade-off for $d_K$: balancing the preservation of relative distances with mitigating the curse of dimensionality. Importantly, we find that keys and queries in the QK space only need to encode location information, without the necessity of carrying complex semantic information like values. This allows $d_K$ to be set to a significantly smaller dimension than $d_V$.
> > W2. "Can the authors clarify the binary expansion? What is the precision of the original d-dimensional vector?"
>
> A2. Thanks for the insightful question.
>
>
> **Clarification of Binary Expansion and Precision** The Z-order curve implementation involves converting the original $d$-dimensional data into a discretized representation with an interleaved bit pattern. Specifically, the original data in each dimension is normalized between 0 and 1 and then discretized to a defined grid size of 4096, which is equivalent to using a 12-bit representation for each coordinate. This means that each element in the $d$-dimensional vector, after normalization, is converted to an integer in the range [0, 4095] with 12 bits of precision.
>
> In this case, if $d = 3$, the final Z-order representation will consist of interleaving the bits from the three 12-bit components, resulting in a total of $3 \times 12 = 36$ bits. This approach reduces the precision requirement compared to a full 32-bit float representation and avoids the potential overflow issue raised in the question. Specifically, instead of expanding to $3 \times 32 = 96$ bits (as with float32), the interleaving is performed on a 36-bit value derived from the original data. This significantly reduces memory overhead and prevents overflow.
>
> **Avoiding overflow and setting precision** Overflow is prevented by discretizing the normalized data to a fixed grid size of 4096. This means each dimension has 12 bits of precision, rather than the 32 bits that would result from using the original float32 representation. The `grid_size = 4096` line specifies the precision with which the data is discretized.
>
> The resulting value, after interleaving bits, will be at most $2^{36}$, which can easily fit within a 64-bit integer without causing an overflow. This precision is sufficient for preserving locality during the Z-order transformation while also avoiding the computational overhead associated with larger bit interleaving.

---

> ### Author Response · Authors · 2024-11-22
> **Response to Reviewer xJi4 (2/3)**
>
> > W3. Can the authors provide a comparison of the time and space usage?
>
> A3. We appreciate this comment and are happy to provide such results, and efficiency results have been added to **Sec 4.6** in the revised version. In particular, we compare with a naive attention implementation available through PyTorch as well as the latest Flash Attention release. We are glad to provide these details to the reviewer. To provide explicit details regarding our implementation, we compare with a naive attention implementation available through PyTorch as well as the latest Flash Attention release. We compare both the time for a single forward pass (used for inference) as well as a forward-backward pass (used for training). Results are provided in milliseconds.
>
> We have provided these in the table below. To provide explicit details regarding our implementation, we compare it with a naive attention implementation available through PyTorch as well as the latest Flash Attention release. We compare both the time for a single forward pass (used for inference) as well as a forward-backward pass (used for training). Time Consumption Results are provided in milliseconds:
>
> | Length | Torch Attention (FWD) | Torch Attention (FWD+BWD) | Mamba (FWD) | Mamba (FWD+BWD) | Flash Attention (FWD) | Flash Attention (FWD+BWD) | ZETA (FWD) | ZETA (FWD+BWD) |
> | ------------- | ------------- | ------------- | ------------- | ------------- | ------------- | ------------- |------------- | ------------- |
> | 4096 | 44.3 | 117.9 | 7.1 | 14.0| 3.4 | 29.2 | 5.6 | 38.2 |
> | 8192 | OOM | OOM |11.8 |23.0|  12.8 | 111.5 | 11.0 | 76.4 |
> | 16384 | OOM  | OOM | 23.5 | 45.7 | 50.4 | 437.7 | 21.7 | 152.6 |
> | 32768 | OOM  | OOM | 47.3 | 91.8 | 198.2 | 1733.5 | 43.0 | 304.8 |
> | 65536 | OOM  | OOM | 94.0 | 183.7 | 805.3 | 7044.1 | 85.8 | 608.2 |
>
> where *OOM* stands for out-of-memory. This in particular demonstrates that our implementation is significantly more efficient than both a naive implementation of attention as well as the latest Flash Attention release. We in particular want to highlight the fact that our method does not suffer from OOM issues while becoming significantly more efficient than Flash Attention as the sequence length increases, validating our claim regarding ZETA theoretical time complexity. Compared to Mamba, ZETA costs slightly less time in forward.
>
> GPU MEM Consumption Results are provided in MB:
> | Length | Torch Attention (FWD) | Torch Attention (FWD+BWD) | Mamba (FWD) | Mamba (FWD+BWD) | Flash Attention (FWD) | Flash Attention (FWD+BWD) | ZETA (FWD) | ZETA (FWD+BWD) |
> | ------------- | ------------- | ------------- | ------------- | ------------- | ------------- | ------------- |------------- | ------------- |
> | 4096 | 17268.1 | 25972.1 | 574.2 | 632.2 | 886.1 | 1784.1 | 1314.1 | 1926.1 |
> | 8192 | OOM | OOM | 904.2 | 1020.2 | 1528.1 | 3324.1 | 2382.1 | 3606.1 |
> | 16384 | OOM | OOM | 1564.2 | 1776.2 | 2812.1 | 6404.1 | 4520.1 | 6968.2 |
> | 32768 | OOM | OOM  | 2884.2 | 3200.2| 5380.1| 12564.1| 8796.2| 13692.2 |
> | 65536 | OOM | OOM |5524.2| 6048.2| 10516.1| 24884.1| 17348.2| 27140.3 |
>
> From this table, ZETA could outperform the torch version attention by a large margin. We admit ZETA needs more memories than flash attention, but in time consumption, ZETA could outperform the latest version of flash attention.
>
> We hope this clarifies the computational efficiency of our method both in terms of time and memory.

---

> ### Author Response · Authors · 2024-11-22
> **Response to Reviewer xJi4 (3/3)**
>
> > Q. How does the use of long context lengths in SOTA LLMs impact the selection of $d_K$ and other hyper-parameters? Will $d_K$ go to 2 or even 1?
>
> A. **LLMs impact the selection of $d_K$** Thank you for your insightful comment. Extending the context length in LLMs to the extreme is indeed a challenging endeavor. Current experiments suggest that setting $d_K = 3$ strikes a good balance, performing well across MQAR, LRA, and WikiText-103. We are actively working on training a larger pre-trained model to verify whether this observation holds consistency for models exceeding 1 billion parameters.
>
> **$d_K$ go to 2 or even 1** We do not set $d_K = 1$ or $2$ in the experiments. Setting $d_K = 3$ has demonstrated its effectiveness in preserving over 60% of correct top-k samples on average. Considering that we typically use 8 attention heads, the probability of overlooking a top-k relevant token in a layer is approximately (1 - 60 %)$^8 \approx $0.07\%. Furthermore, as illustrated in Figure 2(b), performance remains relatively consistent for $d_K \geq 3$, which is further verified in below additional experiments on LRA:
>
> | $d_K$ | 1 | 2 | 3 | 32 | 64 | 128 |
> |:-----:|:-:|:-:|:-:|:--:|:--:|:---:|
> | ListOps | 31.04 | 34.79 | 36.06 | 36.19 | 36.32 | 36.37 |
> | Image | 37.6 | 40.23 | 42.64 | 42.72 | 42.51 | 42.44 |
>
> This demonstrates that $d_K = 3$ is a well-balanced trade-off, which we will continue to verify on larger models. However, in the case of a 2M context length, further validation is necessary, and using more heads can increase the probability of capturing all the ground truth top-$k$ tokens. Notably, the head dimension in models like Llama and Qwen is typically 128. We employ a two-layer neural network to project the input into keys, which helps mitigate the relative distance loss compared to random projections used in the Johnson-Lindenstrauss Lemma.
>
>
>
>
> -----
>
> We appreciate the feedback provided by the reviewer and are hopeful that the responses we provide can help clarify any remaining misunderstandings about our work. We are happy to take their opinion into account towards bettering our work. If the reviewer feels the same, we would be grateful if this could be acknowledged through a revised and improved appraisal of our work.

---

> ### Comment · Reviewer_xJi4 · 2024-11-26
> **Post-rebuttal update**
>
> I thank the authors for their detailed and thoughtful responses during the rebuttal process. After carefully reviewing the clarifications, I believe they significantly enhance the quality and impact of the paper. In light of these improvements, I am convinced that this work merits acceptance, and I have accordingly raised my score.

---

> > ### Author Response · Authors · 2024-11-27
> >
> > We thank you sincerely for your upgraded score. We are very grateful that our response has adequately addressed your prior questions/concerns and that the improvements from this response is visible. We appreciate the time you have spent reviewing our work and are proud to include the changes that have stemmed in no small part from this discussion in our revised manuscript.

---

### Official Review · Reviewer_WjdT · 2024-11-01

**Soundness:** 3
**Presentation:** 3
**Contribution:** 3
**Rating:** 6
**Confidence:** 4

**Summary:**

The authors study an important and timely problem of addressing the quadratic complexity of dot-product attention in Transformers and propose leveraging sparsity by attending to only k most similar keys.

The authors aim to preserve query-key distances ||query_i - key_j|| after projecting queries and keys to a single scalar. To do so, they first reduce the dimension via a random projection per the Johnson-Lindenstrauss lemma.

KNN in d-dim is further reduced to sorting in 1D via the following trick: Let q and k be d-dim vectors. Let the binary representation of the i'th element q[i] be (q_i_31,...q_i_0). Then we can represent q as a number N(q) having a large binary representation (q_0_31,...,q_d-1_31,...q_0_0,...,q_d-1_0). Then |N(q)-N(k)| is likely to preserve the ordering of ||q-k|| as distances are more influenced by more significant bits of each element. This is not always true as q_0_31,...,q_d-1_31,... imposes an ordering on the dimensions whereas ||q-k|| is symmetric. Also note that most data types are limited to 64 bits.

Assuming that this does preserve the pairwise distances of queries and keys, we can sort the key scores N(k1),..,N(kL) and for each q_i use binary search to determine the most similar keys to attend to. The authors point out that for causal LM, we need to exclude future keys from being considered, and propose to use a chunking mechanism where positions are chunked and future key chunks are excluded from consideration.

As the above method is suitable for ||q-k|| and not <q,k>, the authors propose replacing dot-product attention with Cauchy attention where similarity of q and k is computed as 1 / (||q-k||^2 + gamma^2) instead of exp(<q,k> / sqrt(d)). Recent work on linear attention has shown this kernel to be performant.

**Strengths:**

1. The proposed method is interesting, and the trick of projecting to 1D for query-key similarity computation is intuitive.

2. Significant and comprehensive theoretical justification is provided for dimensionality reduction via random projection and is well-supported via detailed proofs.

3. The authors show that this method works well on the multi-query associative recall task that has been used to justify the in-context learning abilities of Transformers. On language modeling on Wikitext-103, the method also works as well as vanilla Transformer.

4. The authors demonstrate that this method outperforms full dot-product attention on Long Range Arena, which is a popular benchmark for stress testing long-range abilities across various modalities.

5. The authors have included a pytorch implementation of their method. I appreciate this and will take it positively into account in my evaluation.

**Weaknesses:**

1. Distances are not preserved even for modest key dimensions, which might be an issue when scaling up. It is not clear if this method can be used as a drop-in replacement for existing pretrained models due to the change in attention similarity.

2. The authors claim that smaller key dimensions do not affect performance. This is not a well-established fact - more comprehensive experiments are required to justify this claim.

3. The chunking part of keys and queries could have been more clearly written, or pseudocode of the method in an appendix would have helped. I looked at the provided torch code in the supplementary but this part looks involved.

4. Benchmarking of speedup / memory usage is not provided.

5. SOTA performance on LRA benchmark is much higher. The performance on the Path-X task is not included, and the authors do not mention this exclusion.

**Questions:**

1. Can you include more details on the chunking part. Also explain where efficiency is coming from.
2. Can you provide resource usage results especially for long-sequences. Also include it for flash attention and other methods such as SSMs which perform better on the tasks that you include the paper.
3. Can you discuss the limitations of your method.
4. Can you provide results on Path-X task in LRA?

~~I am willing to change my score if the authors can address these concerns.~~
**[Post author rebuttal]**: The authors have largely addressed my concerns and I have increased my score.


Typos:
- Equation 6: "(q-Ki)" should be "||q-Ki||"
- Line 232: Period missing after Lemma 3.1

---

> ### Author Response · Authors · 2024-11-22
> **Response to Reviewer WjdT (1/3)**
>
> We appreciate the reviewer for their review that highlights the strengths of our method, such as the ingenuity and intuitiveness of our method. We are happy they appreciate the justification we provide for our method as well as the results we provide to show its ability on various benchmarks. We are also delighted they have made a concerted effort to look at our implementation and personally understand it. In return, we hope the following response we provide can significantly resolve all of their remaining concerns and misunderstandings through additional details and justification that highlight the benefits of our method in a clear and concise manner.
>
> -----
> >S. "Binary representation of Z-order curve imposes an ordering on the dimensions whereas ||q-k|| is symmetric"
>
> A. We admit Z-order curves impose a certain order on the dimensions, but they also ensure that each dimension is represented multiple times in different bit positions. This effectively distributes the influence of each dimension throughout the entire binary representation. As a result, while there is an ordering of dimensions, the interleaving mitigates the dominance of any single dimension, helping to preserve the relative positions of points. Essentially, this bit-level interleaving balances the importance of each dimension and maintains the proximity of points.
>
> The fact that some dimensions may contribute more significant bits does imply that these dimensions could have a larger impact on the ordering. However, for many practical datasets, the interleaving approach used by Z-order curves provides a reasonable approximation of distance preservation. It means that, while the locality may not be perfectly preserved as it would be in the original space, in practice, the Z-order transformation still groups nearby points close together in one-dimensional space, which is sufficient for many use cases such as hierarchical spatial indexing, database queries, and approximate nearest neighbor searches.
>
> >W1. "Distance are not preserved" & W2. "Smaller key dimensions do not affect performance" & "Not clear if this method can be used as a drop-in replacement"
>
> A1 & A2. **Distances are not preserved** & **Smaller key dimensions do not affect performance** We appreciate the reviewer for pointing this out. We believe this may be a minor misunderstanding; while our claim can potentially be misinterpreted in this manner, it is not explicitly what we claim. Our claim is rather that the high dimensionality of the keys and queries is in fact unnecessary. We show this through experimental results on an MQAR task where we significantly reduce the dimensionality of the keys (Section 4.1, first paragraph; Figure 2b) in comparison to standard methods and show that a choice of a small $d_K$ can lead to comparable performance. We further implement the ablation on $d_K$ on Long Range Arena (LRA):
>
> | $d_k$ | 1 | 2 | 3 | 32 | 64 | 128 |
> |:-----:|:-:|:-:|:-:|:--:|:--:|:---:|
> | ListOps | 31.04 | 34.79 | 36.06 | 36.19 | 36.32 | 36.37 |
> | Image | 37.6 | 40.23 | 42.64 | 42.72 | 42.51 | 42.44 |
>
> These experimental results show that $d_k$ will keep a relatively similar performance for $d_k\ge3$ before dropping for $d_k<3$.  Therefore, it further verifies our insight, that keys and queries can carry location information and do not need to carry complex semantic information, and thereby can be set into a smaller value to save computation costs.
>
> **Not clear if ZETA can be used as a drop-in replacement** Thank you for your question. ZETA can not be directly used as a drop-in replacement in pre-trained models because it employs an Euclidean distance metric instead of dot-product similarity and replaces the standard softmax with the Cauchy softmax. As a result, ZETA requires training from scratch.
>
> > W3. "Chunking part of keys and queries could have been more clearly written" & Q1. "More details on the chunking part" & "Where efficiency is coming from"
>
> A3. **Chunking not clearly writing** & **Where efficiency is coming from** We are sorry for the confusion.  The efficiency stems from the fact that we can first project the keys and queries into a 1-D space where a search can be performed. As the space is 1-D dimensional, we know that given a fixed number of tokens $L$, then the resulting time complexity of the sorting is linear (using an efficiently implemented radix sorting operation on z-order curving results). As such, top-$k$ search can be conducted in log-linear time leveraging binary search relative to the sequence length as opposed to quadratic time. Additionally, the use of the top-$k$ tokens for attention computation further reduces the memory complexity as it reduces the number of activations that are computed and stored. We add an illustrative example in Figure 4 and **pseudo-codes** of chunking details in Algorithm 1 L1026-L1049 of **Appendix B** to illustrate this part in the updated pdf version.

---

> ### Author Response · Authors · 2024-11-22
> **Response to Reviewer WjdT (2/3)**
>
> >W4. "Benchmarking of speedup/memory usage is not provided." & Q2 "Resource usage results especially for long-sequences. " & "Including flash attention and other methods such as SSMs"
>
> A4. We have provided these in the table below and efficiency results have been added to **Sec 4.6** in the revised version. To provide explicit details regarding our implementation, we compare it with a naive attention implementation available through PyTorch as well as the latest Flash Attention release. We compare both the time for a single forward pass (used for inference) as well as a forward-backward pass (used for training). Time Consumption Results are provided in milliseconds:
>
> | Length | Torch Attention (FWD) | Torch Attention (FWD+BWD) | Mamba (FWD) | Mamba (FWD+BWD) | Flash Attention (FWD) | Flash Attention (FWD+BWD) | ZETA (FWD) | ZETA (FWD+BWD) |
> | ------------- | ------------- | ------------- | ------------- | ------------- | ------------- | ------------- |------------- | ------------- |
> | 4096 | 44.3 | 117.9 | 7.1 | 14.0| 3.4 | 29.2 | 5.6 | 38.2 |
> | 8192 | OOM | OOM |11.8 |23.0|  12.8 | 111.5 | 11.0 | 76.4 |
> | 16384 | OOM  | OOM | 23.5 | 45.7 | 50.4 | 437.7 | 21.7 | 152.6 |
> | 32768 | OOM  | OOM | 47.3 | 91.8 | 198.2 | 1733.5 | 43.0 | 304.8 |
> | 65536 | OOM  | OOM | 94.0 | 183.7 | 805.3 | 7044.1 | 85.8 | 608.2 |
>
> where *OOM* stands for out-of-memory. This in particular demonstrates that our implementation is significantly more efficient than both a naive implementation of attention as well as the latest Flash Attention release. We in particular want to highlight the fact that our method does not suffer from OOM issues while becoming significantly more efficient than Flash Attention as the sequence length increases, validating our claim regarding ZETA theoretical time complexity. Compared to Mamba, ZETA costs slightly less time in forward.
>
> GPU MEM Consumption Results are provided in MB:
> | Length | Torch Attention (FWD) | Torch Attention (FWD+BWD) | Mamba (FWD) | Mamba (FWD+BWD) | Flash Attention (FWD) | Flash Attention (FWD+BWD) | ZETA (FWD) | ZETA (FWD+BWD) |
> | ------------- | ------------- | ------------- | ------------- | ------------- | ------------- | ------------- |------------- | ------------- |
> | 4096 | 17268.1 | 25972.1 | 574.2 | 632.2 | 886.1 | 1784.1 | 1314.1 | 1926.1 |
> | 8192 | OOM | OOM | 904.2 | 1020.2 | 1528.1 | 3324.1 | 2382.1 | 3606.1 |
> | 16384 | OOM | OOM | 1564.2 | 1776.2 | 2812.1 | 6404.1 | 4520.1 | 6968.2 |
> | 32768 | OOM | OOM  | 2884.2 | 3200.2| 5380.1| 12564.1| 8796.2| 13692.2 |
> | 65536 | OOM | OOM |5524.2| 6048.2| 10516.1| 24884.1| 17348.2| 27140.3 |
>
> From this table, ZETA could outperform the torch version attention by a large margin. We admit ZETA needs more memories than Flash Attention, but in time consumption, ZETA could outperform the latest version of Flash Attention.
>
> We hope this clarifies the computational efficiency of our method both in terms of time and memory.

---

> ### Author Response · Authors · 2024-11-22
> **Response to Reviewer WjdT (3/3)**
>
> > W5. "SOTA performance on LRA benchmark is much higher" &  "Can you provide results on Path-X task in LRA?"
>
> A5. **SOTA performance on LRA:** Thank you for bringing up this point. It is certainly true that some models such as recent state-space models and linear RNNs have demonstrated SOTA performance on the LRA benchmark. However, our work is introducing a novel attention mechanism that is both efficient and scalable. For this reason, our goal is to demonstrate that we improve upon such attention baselines hence our primary comparison is with such methods. We therefore highlight that our method is the best performing attention-based method on this benchmark and we qualify ourselves by never claiming to achieve SOTA performance.
>
> **Path-X:** In the same vein, as existing attention baselines fail to perform above random on Path-X due to lack of inductive bias for such data structures, we exclude it from our results table in order to better demonstrate the improvements that are made, where we follow previous attention variants [1,2]. However, we are actively working on fine-tuning ZETA for Path-X and will release its results in a future revision.
>
> > Q3. "Can you discuss the limitations of your method"
>
> A. **Limitations** Given that our method is a top-$k$ attention mechanism, there are some shared limitations between our method and that of prior work that deals with attention, such as there still potentially being higher chances of ignoring attention to important information (with low attention scores) than full attention methods given the use of only the top-$k$ tokens. We have revised our paper with limitations in **Appendix F**.
>
> [1] The Hedgehog & the Porcupine: Expressive Linear Attentions with Softmax Mimicry, ICLR 2024
>
> [2] FLASHATTENTION: Fast and Memory-Efficient Exact Attention with IO-Awareness, NeurIPS 2022
>
> -----
>
> We hope that our above response is sufficient to alleviate any remaining concerns the reviewer may have regarding our work. We are grateful for their input and believe the points they raised are beneficial towards providing a better understanding of our proposed method. If the reviewer feels the same, we would be greatly appreciative if this could be acknowledged through a revised and improved appraisal of our work.

---

> > ### Comment · Reviewer_WjdT · 2024-11-26
> > **Response to authors**
> >
> > I thank the authors for providing detailed clarifications to my questions.
> > 1. I agree that interleaving the bits from different dimensions would more-or-less remove the asymmetry I referred to earlier.
> > 2. The speedups with Triton are significant and clearly demonstrate Zeta-based top-k to be a viable sparse-attention option.
> > 3. The pseudo-code did not help - however the Figure was helpful.
> > 4. I still think that the experiments section of the paper needs more effort.
> >
> > Overall, I'm still on the fence about this work and not 100% sure about it. However, I appreciate the efforts that the authors have made towards implemeting this and I do think that top-k attention is a promising future direction - how exactly to make it more efficient needs more research.
> >
> > I apologize but I need more time to think about increasing the score. I'll consult with other reviewers and the AC during the reviewer discussion phase and decide accordingly.

---

> ### Author Response · Authors · 2024-12-01
> **Response to Reviewer WjdT**
>
> We are grateful for your response and are happy that some of your initial questions have been addressed. Given that there still appear to be some questions that remain, we are happy to provide additional details in an attempt to resolve these in an attempt to provide even more clarity to our work.
>
> -----
>
> >"Interleaving the bits from different dimensions would more-or-less remove the asymmetry"
>
> We thank you for mentioning this concern. To further address this, we implemented multiple Z-order curve versions. For the $i$-th key with three-dimensional features $K_i = (K_i^1, K_i^2, K_i^3)$, we experimented with different combinations: a 3-Z-Order Curve version involving shifts $(K_i^1, K_i^2, K_i^3), (K_i^2, K_i^3, K_i^1), (K_i^3, K_i^1, K_i^2)$ and a 6-Z-Order Curve version with combinations $(K_i^1, K_i^2, K_i^3), (K_i^1, K_i^3, K_i^2), (K_i^2, K_i^3, K_i^1), (K_i^2, K_i^1, K_i^3), (K_i^3, K_i^1, K_i^2), (K_i^3, K_i^2, K_i^1)$. These multiple Z-Order mappings ensure each key dimension has equal importance, thereby mitigating asymmetry. All implementations collect the same number of top-$k$ tokens. (For the example of 3-Z-order curves, each mapping collects $(k//3)$ tokens.)
>
> | Num of Z-Order Curves| 1 | 3 | 6 |
> |:-----:|:-:|:-:|:-:|
> | ListOps | 42.52 | 42.57 | 42.36 |
> | Image |64.39 | 64.19 | 64.67 |
>
> We observe from these experiments that mitigating the asymmetry of Z-order curves did not lead to significant performance improvements. Accordingly, **this asymmetry is unlikely to have a significant impact on performance empirically**.
>
> > The speedups with Triton are significant and clearly demonstrate Zeta-based top-k to be a viable sparse-attention option.
>
> We sincerely appreciate the praise and are pleased to have provided details that help better highlight the merits of our method.
>
> > The pseudo-code did not help - however the Figure was helpful.
>
> We are glad that the figure was helpful in providing clarity on this front. Given that the figure appears to be more helpful, we will work on incorporating more details in it in order to better walk the reader through our methodology.
>
> > I still think that the experiments section of the paper needs more effort.
>
> We thank the reviewer for their continued correspondence on this front. We **additionally perform zero-shot evaluations** by pretraining a model on the COSMO dataset, a pre-training dataset designed for commonsense inference, and subsequently test its performance on a diverse set of downstream tasks without additional fine-tuning. The evaluation tasks include the Winograd Schema Challenge, ARC-Easy (Arc-E), ARC-Challenge (ARC-C), HellaSwag, PIQA, OpenBookQA (OBQA), BoolQ, and MMLU. The results of the experiments are summarized in the table below:
>
> | Dataset |Winogrd| Arc-E | Arc-C | HellaSWAG | PIQA | OBQA  |BoolQ| MMLU| avg|
> |:-----:|:-:| :-:| :-:| :-:| :-:|:-:|:-:|:-:|:-:|
> |Pythia-125m|50.7 | 34.6| 22.7|29.2 | 60.7 | 28.8 |53.8 |24.6|38.1|
> |Llama-125m| 51.9 | 34.1 | 22.1| 29.4|59.8| 25.4|58.1|24.6|38.2|
> |ZETA-122m|  52.6| 33.5| 23.4 | 27.8| 59.1 | 24.6| 54.0| 24.7| 37.5|
>
> These results demonstrate that ZETA-122m achieves performance comparable to Pythia-125m and LLaMA-125m on the above zero-shot tasks, highlighting its competitiveness across a range of tasks. It also highlights how the attention mechanism in ZETA is sufficient to learn from language data without a significant difference in performacne on real-world evaluation benchmarks. We will update these results in paper in future revisions.
>
> Additionally, we would like to note that we have added some additional experiments throughout the rebuttal period that should answer some of their initial questions. If the reviewer requires any specific experimental results, we would be happy to discuss them further and provide additional results to address these concerns.
>
> -----
>
>
>
> We again thank the reviewer for their continued assessment of our work and willing correspondance throughout this period. We are grateful that they appear to have been moved from our response and are prepared to provide the additional details that are necessary to further convince them to champion our work. If any further questions arise, we are happy to provide answers as soon as possible in the short period that remains.

---

> > ### Comment · Reviewer_WjdT · 2024-12-01
> > **response to authors 2**
> >
> > 1. Thank you for exploring the asymmetry in the dimensions - I agree that this is unlikely to have significant impact on the performance.
> > 2. I appreciate the addition of LM experiments and it is encouraging to see that the results are comparable to standard baselines.
> >
> > I am increasing my score.

---

> > > ### Author Response · Authors · 2024-12-01
> > >
> > > Thank you for your thoughtful feedback and for raising the score of our manuscript. We greatly appreciate your insightful questions, as addressing them has significantly strengthened our paper. We sincerely appreciate the time you have dedicated to reviewing our work and are pleased to incorporate the valuable changes in this discussion into our revised manuscript.

---

### Official Review · Reviewer_pVHs · 2024-11-04

**Soundness:** 4
**Presentation:** 4
**Contribution:** 3
**Rating:** 8
**Confidence:** 3

**Summary:**

Paper is well written, with solid experimental work and theoretical analysis.
Authors introduce Efficient Parallel Top-k Attention by utilising Z-order Curves.  As this implies approximation, thorough experimental evaluation is performed to on several datasets including Long Range Arena, WikiText, MQAR to estimate computation efficiency vs model performance  tradeoffs. On top of this of paper investigate effects of Dimensionality Selection for Key and Query Pairs and Adaptive Cauchy-Softmax Mechanism which help to carefully select tradeoffs.

**Strengths:**

Paper address important area of computation cost (quadratic run time) of classical self-attention, which may be significant for long context. Novel Efficient Parallel Top-k Attention by utilising Z-order Curve was proposed, carefully analysed and evaluated.
On top of this paper this paper investigate dimensionality of keys and queries the trade-off between the curse of dimensionality
and the preservation of relative distances for keys and queries and introduces Adaptive Cauchy-Softmax which allows dynamic adjusting of receptive fields to enhance attention’s flexibility.

**Weaknesses:**

- Evaluation is done on very small scale models (125M), so it could be that they won't translate to large size models.
- While Top-K should take O(N log N) and potentially should be more computation efficient,  it is not clear how much was gained in practice.  Also providing some more details about experiments (what is number of chunks, etc) would be helpful.

**Questions:**

'The insertion position of a query in the key sequence can then be found using a binary search (e.g. with torch.searchsorted), allowing us to retrieve the top-k attended tokens using a window centered on the insertion position'
Could you please clarify this. As far as I understand to find nearest k neighbours,  inspecting several segments of Z-order curve should be necessary or some false positives would be included?


Could you please share computation efficiency measurements (for example average training step time) for baseline transformer and ZETA?

---

> ### Author Response · Authors · 2024-11-22
> **Response to Reviewer pVHs (1/2)**
>
> We are very grateful and would like to thank the reviewer for their positive assessment of our work. We appreciate their highlighting the novelty of our proposed method as well as the through analysis we provide, both experimentally and theoretically. To clarify their remaining questions and potential areas of concern, we hope the following discussion can provide the details necessary.
>
> -----
>
> > W1. "Evaluation is done on very small scale models (125M)"
>
> A1. **Small scale models** Due to the time and resource constraints we have, we only additionally run ZETA with a model size of 239M on WikiText-103:
>
> | Model Size | 124M | 239M |
> |:----------:|:----:|:----:|
> | Perplexity | 26.3 | 22.5 |
>
> Although the results have less finetuning, it still gain lower perplexity. We will continue to train larger scale ZETA model, which will be further released in a public Github repository once the anonymity period concludes.
>
> > W2. "Not clear how much was gained in practice. Also providing some more details about experiments (what is the number of chunks, etc) would be helpful." & Q1. "Could you please clarify (the insertion position of the query in the key sequence)?" & "False positives would be included"
>
> A2. **Details about experiments:** We normally sets the number of chunks to be in $\{4, 8, 16, 32\}$ depending on the sequence length. The hidden dimension is set to $d _ V= 256, 512$ with 8 heads in LRA datasets and $d_v=768$ with 12 heads in WikiText-103. The dimension of keys and queries is set as $d_K=d_Q=3$. We have updated this part in **Appendix C** in a revised version of our manuscript.
>
> **Clarification:** We are happy to address this potential lack of clarity in the description of our method. We hope the following clarification makes this part of the methodology easier to understand.
>
> The efficiency stems from the fact that we can first project the keys and queries into a 1-D space where a search can be performed. As the space is 1-D dimensional, we know that given a fixed number of tokens $L$, then the resulting time complexity of the sorting is linear (using an efficiently implemented radix sorting operation on z-order curving results). As such, top-$k$ search can be conducted in log-linear time leveraging binary search relative to the sequence length as opposed to quadratic time. Since the sequence is already sorted, samples in the window centered at the insertion position are all nearest neighbors (a figure is provided to illustrate this process in the experimental section).  Additionally, the use of the top-$k$ tokens for attention computation further reduces the memory complexity as it reduces the number of activations that are computed and stored.  We add an illustrative example in Figure 4 in **Appendix B** to illustrate this part in the updated pdf version.
>
> **False positives:** While it is possible that some tokens in these windows can be false positives, using multiple heads will result in most top-$k$ tokens being attended to and therefore result in a very negligible change in performance compared to traditional attention.

---

> ### Author Response · Authors · 2024-11-22
> **Response to Reviewer pVHs (2/2)**
>
> >Q2. "Could you please share computation efficiency measurements."
>
> A. We are glad to provide these details to the reviewer and efficiency results have added to **Sec 4.6** in the revised version. To provide explicit details regarding our implementation, we compare with a naive attention implementation available through PyTorch as well as the latest Flash Attention release. We compare both the time for a single forward pass (used for inference) as well as a forward-backward pass (used for training). Results are provided in milliseconds.
>
> We have provided these in the table below. To provide explicit details regarding our implementation, we compare with a naive attention implementation available through PyTorch as well as the latest Flash Attention release. We compare both the time for a single forward pass (used for inference) as well as a forward-backward pass (used for training). Time Consumption Results are provided in milliseconds:
>
> | Length | Torch Attention (FWD) | Torch Attention (FWD+BWD) | Mamba (FWD) | Mamba (FWD+BWD) | Flash Attention (FWD) | Flash Attention (FWD+BWD) | ZETA (FWD) | ZETA (FWD+BWD) |
> | ------------- | ------------- | ------------- | ------------- | ------------- | ------------- | ------------- |------------- | ------------- |
> | 4096 | 44.3 | 117.9 | 7.1 | 14.0| 3.4 | 29.2 | 5.6 | 38.2 |
> | 8192 | OOM | OOM |11.8 |23.0|  12.8 | 111.5 | 11.0 | 76.4 |
> | 16384 | OOM  | OOM | 23.5 | 45.7 | 50.4 | 437.7 | 21.7 | 152.6 |
> | 32768 | OOM  | OOM | 47.3 | 91.8 | 198.2 | 1733.5 | 43.0 | 304.8 |
> | 65536 | OOM  | OOM | 94.0 | 183.7 | 805.3 | 7044.1 | 85.8 | 608.2 |
>
> where *OOM* stands for out-of-memory. This in particular demonstrates that our implementation is significantly more efficient that both a naive implementation of attention as well as the latest Flash Attention release. We in particular want to highlight the fact that our method does not suffer from OOM issues while becoming significantly more efficient than Flash Attention as the sequence length increases, validating our claim regarding ZETA theoretical time complexity. Compared to Mamba, ZETA costs slightly less time in forward.
>
> GPU MEM Consumption Results are provided in MB:
> | Length | Torch Attention (FWD) | Torch Attention (FWD+BWD) | Mamba (FWD) | Mamba (FWD+BWD) | Flash Attention (FWD) | Flash Attention (FWD+BWD) | ZETA (FWD) | ZETA (FWD+BWD) |
> | ------------- | ------------- | ------------- | ------------- | ------------- | ------------- | ------------- |------------- | ------------- |
> | 4096 | 17268.1 | 25972.1 | 574.2 | 632.2 | 886.1 | 1784.1 | 1314.1 | 1926.1 |
> | 8192 | OOM | OOM | 904.2 | 1020.2 | 1528.1 | 3324.1 | 2382.1 | 3606.1 |
> | 16384 | OOM | OOM | 1564.2 | 1776.2 | 2812.1 | 6404.1 | 4520.1 | 6968.2 |
> | 32768 | OOM | OOM  | 2884.2 | 3200.2| 5380.1| 12564.1| 8796.2| 13692.2 |
> | 65536 | OOM | OOM |5524.2| 6048.2| 10516.1| 24884.1| 17348.2| 27140.3 |
>
> From this table, ZETA outperforms the vanilla implementation of attention provided by PyTorch. We admit that ZETA does consume more memory than Flash Attention as of our current implementation which can certainly be improved. However, overall based on this difference as well as the improvements in speed, ZETA remains in our opinion as performant or outperforms the latest version of Flash Attention.
>
> We hope this clarifies the computational efficiency of our method both in terms of time and memory.
>
> -----
>
> We again thank the reviewer for their review which we believe can help improve the presentation of our work. We hope that the response we have provided above is sufficient to better clarify our claims and the benefits of our method. If the reviewer is of the same opinion, we would be grateful for an updated evaluation that can demonstrate this.

---

> > ### Comment · Reviewer_pVHs · 2024-12-02
> >
> > Thank you for conducting those experiments.  Very useful.

---

> > > ### Author Response · Authors · 2024-12-02
> > >
> > > Thank you very much for your valuable feedback. We also sincerely thank you for your constructive insights, which have greatly contributed to improving our paper!

---

### Official Review · Reviewer_36fb · 2024-11-04

**Soundness:** 2
**Presentation:** 3
**Contribution:** 3
**Rating:** 6
**Confidence:** 2

**Summary:**

This paper introduces ZETA, a model leveraging Z-order curves and Adaptive Cauchy-Softmax for efficient top-k attention, specifically aimed at handling long-sequence tasks. The authors propose a novel approach to reduce dimensionality for keys and queries, preserving locality while enabling faster token retrieval through Z-order-based projections. Their experimental results demonstrate competitive or superior performance on various benchmarks, suggesting ZETA’s potential as an efficient alternative for long-range attention tasks.

**Strengths:**

Originality : The paper introduces a novel use of Z-order curves for dimensionality reduction in attention mechanisms, adding a fresh approach to efficient top-k attention.
Quality:  Methodologically sound with theoretical insights like the Johnson–Lindenstrauss Lemma for dimensionality reduction. Experimental results across benchmarks support the model’s performance, though more tests could strengthen findings.
Clarity : The paper is well-organized and accessible, with clear explanations of techniques like Adaptive Cauchy-Softmax and Z-order projection, aiding reader comprehension.
Significance : ZETA shows potential in handling long sequences more efficiently, which could benefit applications in NLP and large-scale modeling. Further testing on diverse tasks would solidify its impact.

**Weaknesses:**

bout the experiment about  d_k​ in Experiments:
The authors test a narrow range of dkd_kdk​ values (1, 2, 3, 8, and 32), which may not reflect real-world applications where larger dimensions are common. Why weren’t higher dkd_kdk​ values explored? Expanding this range, particularly with larger model dimensions, could yield more generalizable insights.

Unclear Transferability of \gamma in Adaptive Cauchy-Softmax:
The transferability of the trainable parameter γ\gammaγ in Adaptive Cauchy-Softmax across different tasks or datasets is unclear. Could the authors offer evidence or discussion on its robustness beyond the tested contexts?
Lack of Formal Justification for Euclidean Distance Claim:
The claim that Euclidean distance is more effective for low-dimensional top-k methods lacks formal support. Providing a proof or theoretical justification would enhance the validity of this argument.

**Questions:**

Why choose ASSOCIATIVE RECALL in section 4.2 ?
Testing different d_k values on more Long Range Arena (LRA) tasks would provide a fuller picture of task-specific impacts on dimensionality selection. Could the authors evaluate dkd_kdk​ across a wider range of LRA tasks to strengthen their findings?

---

> ### Author Response · Authors · 2024-11-22
> **Response to Reviewer 36fb (1/2)**
>
> We would like to thank the reviewer for the time and effort they have placed towards providing a thorough evaluation of our work. We are delighted that they mention the originality, quality, clarity as well as the potential downstream significance of the method we propose. We are hopeful that the responses we provide to the raised weaknesses and questions below can sufficiently address their remaining concerns regarding our work as well as provide the necessary details to clarify possible misunderstandings that might have resulted from their absence.
>
> -----
>
> > W1. "Narrow range of $d_K$ in Experiments" & "Why weren’t higher $d_K$ values explored?" & Q1. "Testing different $d_K$ values on more Long Range Arena (LRA)" & "Why choose Associative Recall "
>
> A1: **Range  of $d_K$ & Why not higher values:** Thank you for raising this point. We initially test a smaller range of $d_K$ values in order to demonstrate that lower values are in fact sufficient to properly replicate the effects of standard attention. Our values are meant to show that by choosing a value that is an order of magnitude smaller than what is commonly used, performance still remains comparable.
>
>
> In the Multi-Query Associative Recall (MQAR) datasets, performance remains consistent for $d_K \ge 3$, with no noticeable improvement beyond this value. Therefore, results for $d_K \ge 64$ are not included in the tables presented in our paper. Additionally, in this paper, $d_K$ refers to the dimension of key in each head, which is normally within the range $d_K\in\{1,2,3,8,32,64,128\}$. For both Llama and Qwen models, the head dimension is typically 128, resulting in a total key dimension of 4096 across 32 heads in each layer. As pointed out above, we did not test $d_K = 64$ or $128$ because performance does not improve with $d_K$ values larger than 32.
>
>
> **Testing on LRA** We are happy to include additional ablations on the value of $d_K$ to demonstrate that across different tasks, the range of values that are used is sufficient. We further expand the ablation study presented in Figure 2(b) on ListOps and Images in LRA, examining attention performance with varying $d_K$, as detailed in the table below:
>
> | $d_K$ | 1 | 2 | 3 | 32 | 64 | 128 |
> |:-----:|:-:|:-:|:-:|:--:|:--:|:---:|
> | ListOps | 31.04 | 34.79 | 36.06 | 36.19 | 36.32 | 36.37 |
> | Image | 37.6 | 40.23 | 42.64 | 42.72 | 42.51 | 42.44 |
>
> The experimental results indicate that $d_K$ maintains relatively consistent performance for $d_K \geq 3$, but performance declines for $d_K < 3$. This further supports our insight that, unlike values, keys and queries primarily encode location information rather than complex semantic information. Therefore, setting $d_K$ to a small enough value can effectively reduce computation costs without significant performance loss. This motivates our choice of $d_K$ values that were used for our experiments.
>
>
> **Why choose Associative Recall** Thank you for asking. We choose to conduct experiments with Associative Recall as it is a synthetic task that is specifically meant to test the ability to retain information across a sequence, which can be particularly daunting in the case of long sequences, such as [1,2,4]. Given that the use of top-k attention is aimed towards improving attention efficiency without significant performance drops, this task is a good benchmark for the effectiveness of our method in terms of the general ability to retain and retrieve information across long distances (within a sequence) and serves as an indicator of possible strong performance in other long-context tasks. However, it is certainly true that such a task might not be fully representative of real world tasks, hence our initial submission also included experiments on tasks such as LRA and language modeling.

---

> ### Author Response · Authors · 2024-11-22
> **Response to Reviewer 36fb (2/2)**
>
> >W2. "Transferability of $\gamma$" & "Discussion on its robustness"
>
> A2. **Transferability** Verifying transferability through current datasets such as MQAR, LRA, and WikiText-103 is not suitable. MQAR and WikiText-103 are not directly related to transfer tasks, and LRA consists of entirely different tasks, including Images and ListOps, which may lead to negative transfer for the trained model. Zero-shot language modeling tasks are more suitable for evaluating transferability and robustness, and we are currently working on this. However, the complexity of the implementation of the zero-shot benchmark is significant, and we will update our findings if results are obtained before the end of the discussion period. Otherwise, they will be included in future revisions.
>
> **Discussion on its robustness**  The trainable parameter $\gamma$ in Cauchy Softmax can be viewed as a form of Temperature Scaling, which is effective at controlling the contextual relevance between queries and keys, as confirmed by the most recent work [5]. In our paper, we also highlight that a trainable $\gamma$ helps prevent entropy collapse or explosion, thereby adaptively balancing attention across sequences.
>
> We further provide the ablation on the trainable $\gamma$ by replacing it with a constant 1, and the cauchy distribution turns into a static t-student distribution.
>
> |         | $\gamma$=1 | Trainable $\gamma$ | Improvement |
> |:-------:|:----------:|:------------------:|:-----------:|
> | ListOps | 41.99 | 42.52 | 0.53 &uarr; |
> | Image | 64.19 | 64.39 | 0.20 &uarr; |
>
> A trainable $\gamma$ consistently outperforms a fixed $\gamma = 1$. In ZETA, $\gamma$ can be fine-tuned to optimize performance depending on the dataset and the model's depth, providing additional flexibility and improving robustness.
>
> >W3. **Justification for Euclidean Distance Claim**
>
> A3. We use Euclidean distance as it is well-suited for the k-NN method in identifying the top-k attended tokens. In contrast, dot-product similarity cannot be directly applied to k-NN searches for top-k tokens without normalization, as highlighted in [3]. Experimental results further show that a normalized dot product performs worse than Euclidean distance, as demonstrated in additional MQAR experiments below:
> | $d_k$ | 1 | 2 | 3 | 4 |
> |:-----:|:-:|:-:|:-:|:-:|
> | Negative Euclidean | 64.6 | 99.4 | 99.4 | 99.3 |
> | Inverse Euclidean | 22.9 | 81.3 | 99.6 | 99.9 |
> | Cauchy Softmax | 74.5 | 99.6 | 99.5 | 99.2 |
> | Normalized Dot Prod | 62.6 | 92.6 | 99.3 | 99.1 |
>
> Specifically, Euclidean distance-based methods, such as Negative Euclidean with traditional softmax and Cauchy Softmax, outperform the dot product for top-k attention when using small $d_k \le 4$, as adopted in ZETA.
>
> Figure 1 in the paper illustrates an example with $d_K = 1$, where relying on dot products to select top-k tokens or identify k-nearest neighbors leads to incorrect predictions. While theoretical justifications for using Euclidean distance are currently beyond the scope of this rebuttal due to time constraints, we are committed to further investigating this and will provide updates in a future version.
>
> -----
>
> ### References
>
> [1] The Hedgehog & the Porcupine: Expressive Linear Attentions with Softmax Mimicry, ICLR 2024
>
> [2]  Transformers are SSMs: Generalized Models and Efficient Algorithms Through Structured State Space Duality, ICML 2024
>
> [3] IceFormer: Accelerated Inference with Long-Sequence Transformers on CPUs, ICLR 2024
>
> [4] Zoology: Measuring and Improving Recall in Efficient Language Models, ICLR 2024
>
> [5] Selective Attention: Enhancing Transformer through Principled Context Control, NeurIPS 2024
>
> -----
>
> We again thank the reviewer for spending their time to evaluate our work; we are immensely grateful for their feedback and suggestions which we believe will go a long way in improving the clarity of our work and better justifying claims that could benefit from improved presentation. We are hopeful that this response we have provided has the ability to demonstrate this and would be appreciative if this could be further reflected in an updated evaluation taking these factors into account.

---

> > ### Comment · Reviewer_36fb · 2024-12-03
> >
> > Thank you for providing such comprehensive data. This complete analysis has addressed all my concerns.

---

> ### Author Response · Authors · 2024-12-01
> **Supplementary Response on the Transferability of $\gamma$**
>
> > Transferability of $\gamma$
>
> **Supplementary response to Transferability of $\gamma$** Zero-shot language modeling tasks serve as a suitable benchmark for evaluating the transferability of $\gamma$. To further assess this, we perform zero-shot evaluations by first pre-training a model on the COSMO dataset, a dataset specifically designed for commonsense inference. Subsequently, we test its performance on a diverse set of downstream tasks without any additional fine-tuning. These tasks include the Winograd Schema Challenge, ARC-Easy, ARC-Challenge (ARC-C), HellaSwag, PIQA, OpenBookQA (OBQA), BoolQ, and MMLU.
>
> The experimental results are summarized in the table below:
>
> | Dataset |Winogrd| Arc-E | Arc-C | HellaSWAG | PIQA | OBQA  |BoolQ| MMLU| avg|
> |:-----:|:-:| :-:| :-:| :-:| :-:|:-:|:-:|:-:|:-:|
> |Pythia-125m|50.7 | 34.6| 22.7|29.2 | 60.7 | 28.8 |53.8 |24.6|38.1|
> |Llama-125m| 51.9 | 34.1 | 22.1| 29.4|59.8| 25.4|58.1|24.6|38.2|
> |ZETA-122m|  52.6| 33.5| 23.4 | 27.8| 59.1 | 24.6| 54.0| 24.7| 37.5|
>
> These results demonstrate that trained $\gamma$ in ZETA-122m **exhibits good transferability and robustness and achieves competitive performance** across a variety of NLP tasks. Updated results and further refinements on these experiments will be included in subsequent revisions of the paper.
>
> ----------------------------------
>
> We sincerely thank the reviewer for their valuable feedback. As the discussion phase approaches its conclusion, we would greatly appreciate the opportunity to engage in further discussion with you. Our aim is to ensure that our responses fully address your concerns and to explore any additional questions or comments you may have.

---

> ### Author Response · Authors · 2024-12-03
> **Response to Reviewer 36fb**
>
> Thank you so much for your response. We are delighted to know that the analysis we provided was helpful in addressing all your concerns. If it aligns with your satisfaction, we would deeply appreciate it if you could consider revising your score to better reflect your positive feedback.  Once again, thank you for your time and consideration and we are immensely grateful for your insights that have greatly contributed to improving our paper.

---

### Author Response · Authors · 2024-11-22
**General Response + Revision Summary**

We gratefully acknowledge all the reviewers for their insightful comments. We are happy that the reviewers describe our work as **interesting/original** [Reviewers 36fb, pVHs, WjdT], **significant/important** [Reviewers 36fb, pVHs, WjdT], **methodologically and theoretically sound** [Reviewers 36fb, pVHs, WjdT, xJi4]. We also appreciate that most found our work **clear and well-written** [Reviewers 36fb, pVHs, xJi4]. We would also like to specifically thank Reviewer WjdT for acknowledging our theoretical proofs as well as making the added effort to understand the code implementation that was attached.

To summarize our revisions and rebuttals, we have

- We have included tables (provided below) measuring speed (in milliseconds) and memory consumption of our implementation in rebutalls and a revised version of our manuscript in a new **Section 4.6**. (To pVHs & WjdT & xJi4)
- Provided detailed explanations of our **Triton implementation of ZETA** in **Appendix D and E**, including both the forward pass and the gradient derivations for the backward pass of effective top-$k$ indexing and Cauchy softmax. Additionally, include the Triton-based version of ZETA in the supplementary materials.
- Provided additional insight into the choice of $d_k$ through additional experiments in **Appendix G.1**. (To 36fb & WjdT & xJi4)
- Provided justification for design choices, such as our Adaptive Cauchy Softmax in **Appendix G.2**. (To 36fb).
- Clarified points that some reviewers found unclear, such as justifying our claim regarding Euclidean Distance (To 36fb) and clarifying our methodology such as the chunking mechanism, which is also added an illustrative example in **Appendix B** (to pVHs & WjdT).


Furthermore, while we have addressed the following point in individual rebuttals, we would like to again acknowledge the common question regarding the efficiency of our method. As this was a common point raised by all reviewers, we want to acknowledge that we have taken this important point into account and wish to again provide these results in this general response.

| Length | Torch Attention (FWD) | Torch Attention (FWD+BWD) | Mamba (FWD) | Mamba (FWD+BWD) | Flash Attention (FWD) | Flash Attention (FWD+BWD) | ZETA (FWD) | ZETA (FWD+BWD) |
| ------------- | ------------- | ------------- | ------------- | ------------- | ------------- | ------------- |------------- | ------------- |
| 4096 | 44.3 | 117.9 | 7.1 | 14.0| 3.4 | 29.2 | 5.6 | 38.2 |
| 8192 | OOM | OOM |11.8 |23.0|  12.8 | 111.5 | 11.0 | 76.4 |
| 16384 | OOM  | OOM | 23.5 | 45.7 | 50.4 | 437.7 | 21.7 | 152.6 |
| 32768 | OOM  | OOM | 47.3 | 91.8 | 198.2 | 1733.5 | 43.0 | 304.8 |
| 65536 | OOM  | OOM | 94.0 | 183.7 | 805.3 | 7044.1 | 85.8 | 608.2 |

where *OOM* stands for out-of-memory. This in particular demonstrates that our implementation is significantly more efficient that both a naive implementation of attention as well as the latest Flash Attention release. We in particular want to highlight the fact that our method does not suffer from OOM issues while becoming significantly more efficient than Flash Attention as the sequence length increases, validating our claim regarding ZETA theoretical time complexity. Compared to Mamba, ZETA costs slightly less time in forward.

GPU MEM Consumption Results are provided in MB:
| Length | Torch Attention (FWD) | Torch Attention (FWD+BWD) | Mamba (FWD) | Mamba (FWD+BWD) | Flash Attention (FWD) | Flash Attention (FWD+BWD) | ZETA (FWD) | ZETA (FWD+BWD) |
| ------------- | ------------- | ------------- | ------------- | ------------- | ------------- | ------------- |------------- | ------------- |
| 4096 | 17268.1 | 25972.1 | 574.2 | 632.2 | 886.1 | 1784.1 | 1314.1 | 1926.1 |
| 8192 | OOM | OOM | 904.2 | 1020.2 | 1528.1 | 3324.1 | 2382.1 | 3606.1 |
| 16384 | OOM | OOM | 1564.2 | 1776.2 | 2812.1 | 6404.1 | 4520.1 | 6968.2 |
| 32768 | OOM | OOM  | 2884.2 | 3200.2| 5380.1| 12564.1| 8796.2| 13692.2 |
| 65536 | OOM | OOM |5524.2| 6048.2| 10516.1| 24884.1| 17348.2| 27140.3 |

From this table, ZETA outperforms the standard PyTorch implementation of attention by a large margin. Admittedly, ZETA consumes slightly more memory than the latest Flash Attention, but in totality we believe our results here demonstrate ZETA to be as good if not better than Flash Attention.

We would also acknowledge that significant improvements are likely further possible for ZETA as we (the authors) are not as experienced with system-level programming such as CUDA and Triton and therefore our implementation is likely not as mature as a library such as Flash Attention. However, we would like to highlight our efforts in making our implementation as strong as possible and hope that this table is sufficient to leave the reviewers with the same impression, which we hope can be taken into consideration.

---

### Meta-Review · Area_Chair_69f3 · 2024-12-20

**Metareview:**

The paper proposes ZETA, a novel method for efficient parallel top-k attention using Z-order curves to successfully reduce computational complexity, enabling better scaling to long sequences while maintaining or improving performance compared to the standard attention mechanism.
Reviewers were consistently positive about the paper's technical contributions and extensive empirical validation, noting clear innovations in efficient attention, giving progress in an area of high interest and relevance for recent AI models.
All 4 reviewers recommend acceptance.

**Additional Comments On Reviewer Discussion:**

The author feedback phase was productive, converging to good agreement between all parties

---

### Decision · Program_Chairs · 2025-01-22

Accept (Poster)